

# CLIMFILL: A Framework for Intelligently Gap-filling Earth Observations

Verena Bessenbacher[1], Sonia I. Seneviratne[1], and Lukas Gudmundsson[1]

[1]ETH Zürich, Rämistrasse 101, 8092 Zürich, Switzerland

**Correspondence:** Verena Bessenbacher (verena.bessenbacher@env.ethz.ch)

**Abstract.** Earth observations have many missing values. Their abundance and often complex patterns can be a barrier for combining different observational datasets and may cause biased estimates. To overcome this, missing values in geoscientific data are regularly infilled with estimates through univariate gap-filling techniques such as spatio-temporal interpolation. However, these mostly ignore valuable information that may be present in other dependent observed variables. Here we propose CLIMFILL, a multivariate gap-filling procedure that builds up upon simple interpolation by additionally applying a statistical imputation method which is designed to account for dependence across variables. In contrast to popular up-scaling approaches, CLIMFILL does not need a gap-free gridded "donor" variable for gap-filling. CLIMFILL is tested using gap-free ERA5 reanalysis data of ground temperature, surface layer soil moisture, precipitation and terrestrial water storage to represent central interactions between soil moisture and climate. These observations were matched with corresponding remote sensing observations and masked where the observations have missing values. CLIMFILL successfully recovers the dependence structure among the variables across all land cover types and altitudes, thereby enabling subsequent mechanistic interpretations. Soil moisture-temperature feedback, which is underestimated in high latitude regions due to sparse satellite coverage, is adequately represented in the multivariate gap-filling. Univariate performance metrics such as correlation and bias are improved compared to spatiotemporal interpolation gap-fill for a wide range of missing values and missingness patterns. Especially estimates for surface layer soil moisture profit taking into account the multivariate dependence structure of the data. The framework allows tailoring the gap-filling process to different environmental conditions, domains or specific use cases and hence can be used as a flexible tool for gap-filling a large range of remote sensing and in situ observations commonly used in climate and environmental research.

## 1 Introduction

### 1.1 Missing observations in Earth system science

Observing the Earth surface from the ground or space is an endeavour that has significantly contributed to advance our understanding of the Earth system and has played a vital role in the fields of data assimilation (Bauer et al., 2015), global freshwater



hydrology (Lettenmaier et al., 2015), Earth surface modeling (Balsamo et al., 2018) and the study of climate extremes in the
land-atmosphere system (Dorigo et al., 2017).

A plethora of instruments observes variables relevant for determining the state of the Earth at any given time. However, the
observational record is highly fragmented: Available observational datasets differ in spatio-temporal resolution, frequency or
extent and have different patterns of missing values. For example, ground observations such as weather stations (e.g. Harris
et al. 2020b, Lawrimore et al. 2011) and FLUXNET towers (Pastorello et al., 2020) give an intricate view of a range of vari-
ables at high temporal resolution, but are unevenly scattered across the globe. Remote sensing observations from space have a
extensive spatial coverage, but suffer from inhomogeneities, missing values and measurement limitations (Lettenmaier et al.,
2015; Shen et al., 2015; de Jeu et al., 2008). As a consequence, the observational record suffers from complex, large-scale
and unavoidable missing values that hinder further analysis and can obscure physically consistency among variables. How-
ever, combining observations from several physical variables into a coherent "view" of the state of the Earth system is crucial
for many applications. These include, but are not limited to, analysis of local and regional land surface dynamics, tracing of
compound extreme events or observational water and energy budget closures. The necessity of creating a global, physically
coherent observational dataset of the Earth's state is also highlighted through international initiatives such as the Digital Twin
Earth Initiative from ESA (Bauer et al., 2021).


Combining observations or derived data products is often hindered by their different underlying assumptions, different
spatio-temporal extent and resolution as well as different patterns of missing values. Several gridded observational products
attempt to overcome this fragmentation by combining one or more observations to a spatially or temporally complete dataset
(Brocca et al., 2014; Huffmann et al., 2019) or estimate variables that are only observed through sparse station networks
through statistical up-scaling (Gudmundsson and Seneviratne, 2015; Martens et al., 2017; Jung et al., 2011, 2009). These
gridded observations have different model assumptions, and usually scale somewhere between geostatistical approaches like
interpolation (Mariethoz et al. 2012; Haylock et al. 2008, for an overview see Shen et al. 2015) and a mixture of sophisticated
machine learning and mechanistic models (Gudmundsson and Seneviratne, 2015; Alemohammad et al., 2017; Jung et al., 2009;
Ghiggi et al., 2019; Kadow et al., 2020; Tramontana et al., 2016).


Within the field of land-climate dynamics, the fragmentation of the observational record is particularly apparent. At the
land-atmosphere boundary a complex interplay between soil moisture, temperature and precipitation governs much of the wa-
ter and energy balance at the surface (Seneviratne et al., 2010). The entirety of atmospheric and terrestrial processes influences
local climate (Seneviratne et al., 2010; Greve et al., 2014), the development of hot and dry extreme events (Miralles et al.,
2019; Mueller and Seneviratne, 2012), freshwater availability (Gudmundsson et al., 2021) and climate change (Seneviratne
et al., 2010). These interactions are inherently multivariate and act on different timescales, making it necessary to observe the
variables at a fine resolution to detect feedbacks and mechanisms. Consequently, the study of land-climate dynamics requires
observations spanning several components of the Earth system, including the land water and energy balances as well as the the



**Table 1.** Variables relevant for land climate interactions and corresponding observational datasets, sorted after the scientific domain they are mostly used in. Note that some observational products are not global, but cover only a larger region (e.g. E-OBS only covers Europe). For the references for each of the products, see Supplementary Table A2

| domain | variable | satellite observation | | in situ observation | gridded observation | |
| --- | --- | --- | --- | --- | --- | --- |
| | | orbiting | geostationary | | product name | gridding technique |
| atmosphere | 2-meter temperature | – | – | SYNOP stations FLUXNET stations | E-OBS, CRU | interpolation |
| | precipitation | GPM | – | SYNOP stations FLUXNET stations | E-OBS, CRU | interpolation |
| land water | surface soil moisture | ESA-CCI-SM | – | ISMN | – | – |
| | root zone soil moisture | – | – | ISMN | – | – |
| | terrestrial water storage | GRACE | – | – | – | – |
| | evapotranspiration | – | – | FLUXNET | WECANN | neural network |
| | | | | | GLEAM | neural network |
| | runoff | – | – | GSIM | G-RUN | ensemble of machine learning techniques |
| land energy | latent heat | | | (see evapotranspiration) | | |
| | sensible heat | – | – | FLUXNET stations | WECANN | neural network |
| | longwave radiation | CERES | – | FLUXNET stations | – | – |
| | shortwave radiation | CERES | – | FLUXNET stations | – | – |
| | ground heat flux | – | – | FLUXNET stations | – | – |
| | ground temperature | MODIS | SEVIRI | FLUXNET stations | – | – |

atmospheric state.


In Table 1, we show an example of the fragmented world of Earth observations that challenge investigations in land-climate dynamics. The table highlights two issues that are typically encountered when analysing the observational record of the Earth system: there is either none or more than one observation system available for each variable. For example, evaporation is a key variable linking the water and the energy cycle at the surface, but it cannot be observed from space and is only sparsely mea-

sured on ground based observatories (Martens et al., 2017). If there is more than one observation of relevant variables, those are usually difficult to combine because of inherently different measurement procedures. An example for this is temperature. Space-borne observations see the temperature of the Earth surface, while in situ stations typically measure temperature in the atmosphere at two meters height. Combining those products can lead to errors in the estimate of surface energy partitioning (Balsamo et al., 2018) or might lead to diverging results when attempting model evaluation. Soil moisture is affected by both

issues: While soil moisture can be observed from space, the microwave signal only penetrates the few first centimeters of the soil (Dorigo et al., 2017). Consequently, information on vegetation-available root zone water which is central to many land-atmosphere coupling effects is only available from sparse in situ observations (Dorigo et al., 2017), whilst surface soil moisture





is measured both from space and in situ. Terrestrial water storage is available globally from the GRACE satellite (Swenson, 2012), but tracks all water on land, including soil moisture, ground water and lake water. Hence we have several datasets for soil moisture that are difficult to combine.

Coming from the realm of physical modeling, reanalysis can be viewed as another class of gap-free reconstructions of the state of the Earth system, and are often the default dataset for a range of applications (Hersbach et al., 2020; Dee et al., 2011; Gelaro et al., 2017). Atmospheric reanalysis typically assimilates a wide range of observations into global weather models. However, reanalysis products are by construction model-driven. They are therefore subject to model biases (Bocquet et al., 2019) and issues with model independence can arise if classical reanalysis products are used for model validation. Moreover, the observational record of the Earths surface is generally underutilised in state-of-the-art reanalysis products. The large fraction of missing values is cited as one of the mentioned reasons for this shortcoming (Dorigo et al., 2017). For example, in state-of-the-art atmospheric reanalysis product ERA5 the already difficult observational record of soil moisture is used only sparsely (Hersbach et al., 2020), although the added value for example remote sensing soil moisture assimilation been shown for weather forecast models (Zhan et al., 2016) and flood forecasting (Brocca et al., 2014; Sahoo et al., 2013). Incomplete observation assimilation can therefore lower forecast accuracy and for example have consequences on the prediction of extreme events. A gap-filling procedure that can combine different observations into a coherent gap-free dataset could be used as a possible pre-processing step in reanalysis to enable a more thorough usage of available land observations.

Consequently, Balsamo et al. (2018) note the need for more multivariate Earth observation datasets apart from reanalysis. At the same time, Bauer et al. (2021) mention an ongoing trend to reshape classical reanalysis such that physical modeling and fragmented observation can be harmonised into a combined product by the use of machine learning techniques wherever processes are unknown or difficult to parameterise. In the following, we present an approach to consolidate fragmented Earth observations into a coherent, multivariate, gap-free dataset by tackling the problem of missing values in the multivariate Earth observation record with the gap-filling framework CLIMFILL. Distinguishing the approach from reanalysis, we do not aim to assimilate observations with a pre-defined physical model, but to leverage the power of modern statistical techniques to produce dependable and physically consistent estimates of essential Earth system observations. The newly developed methodology is exemplarily tested for variables relevant in the study of land-atmosphere dynamics.

## 1.2 A brief review of gap-filling methods

### 1.2.1 Gap-filling in the methodological literature

The methodological literature offers a theoretical framework for the problem of missing values in any kind of data (Rubin, 1976). Typically, the simplest form of gap management is referred to as list-wise deletion, where only data points are considered if all variables are observed. However, this approach can lead to very large data loss. Furthermore, statistics derived from incomplete data can be biased if the data are missing not at random (Rubin, 1976). Consequently, the pattern in which the

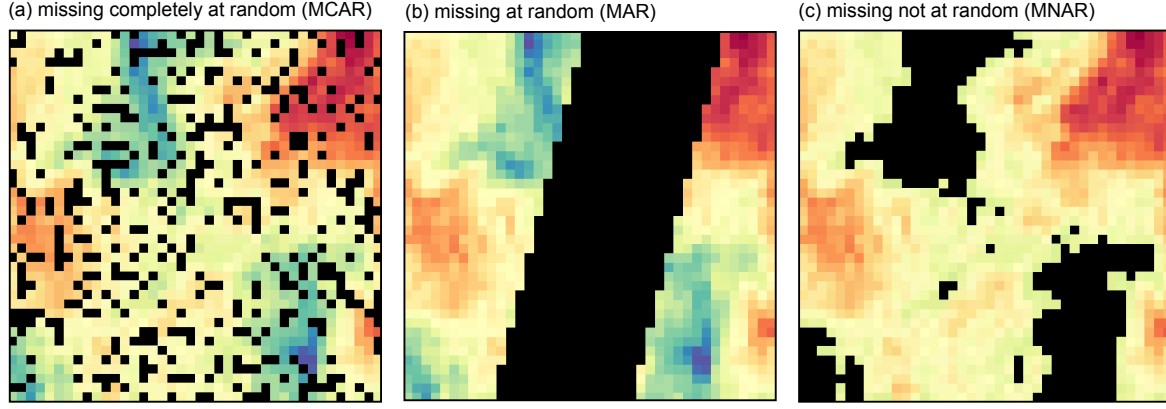

**Figure 1.** Examples of the three patterns in which values can be missing: (a) Missing completely at random (MCAR), (b) Missing at random (MAR) and (c) Missing not at random (MNAR). The MCAR missingness is created by setting randomly drawn grid points to be missing. For MAR missingness, a patch of the data was removed to mimic satellite swaths. In MNAR missingness, all values below a certain threshold are missing.

data are missing (i.e., the "missingness") is one of the most important factors when estimating the impact of missing values (Little and Rubin, 2014). In particular, Rubin (1976) categorizes three ways in which data can be missing: missing completely at random (MCAR), missing at random (MAR) and missing not at random (MNAR). All these three missingness patterns can be observed in Earth observation data:

– If the probability that a data point is missing is not dependent of any process, the missingness is described as missing completely at random (MCAR, Figure 1 a). This is rarely the case in Earth observations.

    – Satellite data are often missing because of satellite swaths. For example orbiting satellites, e.g. measuring soil moisture with a microwave sensor, do not pass certain regions at certain times (Figure 1 b). Here, the fact that we can't measure the soil moisture at a certain space-time-point is not dependent on the actual soil moisture at this point. In other words, 115     the soil moisture is not significantly lower or higher where the satellite does not pass through. Therefore, the probability of a data point being missing is not dependent on the value of the missing data point. Rubin (1976) call this missingness pattern missing at random (MAR).

    – The most complex missingness pattern is missing not at random (MNAR). Here, the mechanism that obscures data points depends on the data that are missing. This mechanism can either be a function of the observed variables, for 120     example when values above or below a certain threshold are not observable. Or if missingness is controlled by a different, but unobservable variable. In the case of an exemplary satellite measuring soil moisture via microwave retrievals, the measurement over dense vegetation represents more the water content of the canopy rather than the one of the soil. Hence the data at such points are masked during post-processing, leading to large patches of missing values especially in





tropical forests. Here, we cannot safely assume that the soil moisture below dense vegetation is not significantly different
from the soil moisture that is not missing. Therefore, we cannot assume independence between the fact that a point is
missing and the unobserved value of the missing point. We observe MNAR missingness (Figure 1 c).

Geoscientific data are in a large part missing not at random (MNAR), making statistical measures of the data biased (van
Buuren, 2018) and gap-filling challenging (see for example Cowtan and Way (2014)). Ghahramani and Jordan (1994) show
that gap-filling with the help of statistical tools (called statistical imputation) of missing data is possible for MCAR and MAR
in both a Bayesian and a Maximum Likelihood setting, but note that MNAR data cannot be tackled with the same methods.
However, imputation can still be successful if a high degree of dependence between MNAR variables increases their mutual
information. We argue that this is especially the case for geoscientific observations, since the variables are often directly linked
through a number of processes.

A wide range of algorithms that make use of cross-variable dependence to estimate missing values exist in statistical liter-
ature. Gaussian Processes are a natural choice for gap-filling problems, but they have limitations when moving to large data
(Heaton et al., 2019). Other approaches center around low-rank matrix recovery or eigenvalue analysis for estimating miss-
ing values (Davenport and Romberg, 2016; Mazumder et al., 2010). Iterative procedures like the MICE-Algorithm ("Multiple
imputation by chained equation", van Buuren (2018)) are suited well for multivariate imputation and scale to large data, but
cannot account for neighborhood relations. Regression-based multivariate gap-filling algorithms like these have, to the best of
our knowledge, not yet been applied in the geoscientific context.

### 1.2.2   Gap-filling in Earth system sciences

Gap-filling is common for Earth observation data. It is used to fill gaps originating from sensor failure or sensor limitations (Liu
et al., 2018; Shen and Zhang, 2009; Pastorello et al., 2020), to extrapolate into undersampled regions (Jung et al., 2011, 2009;
Cowtan and Way, 2014; Gudmundsson and Seneviratne, 2015; Ghiggi et al., 2019) or to get estimates for regions obscured to
the sensor by clouds, dense vegetation, flight geometry or other influences (Brooks et al., 2012; Zeng et al., 2015; Shen and
Zhang, 2009; Huffmann et al., 2019).

These gap-filling methods can be categorized along the data dimension used for producing estimates. For example, a classi-
cal method for gap-filling time series and spatial data is interpolation, e.g. in the form of Kriging. There is a growing body of
literature of different methods that are originally equipped with dealing with only spatial or temporal relations are expanded
and altered to take into account the information from the other dimension as well (von Buttlar et al., 2014; Gerber et al., 2018).
However, these gap-filling methods are univariate and cannot account for information provided by another variables.

Recent literature offers new approaches that translate existing gap-filling methods into the multivariate setting. Temporal
methods for gap-filling of point-scale data are extended to account for other variables (Moffat et al., 2007; Liu et al., 2018), but





**Table 2.** Main CLIMFILL settings per step and method class. Each task can be performed using other method of the corresponding method class.

| Step | Task | CLIMFILL-RF (this study) | Examples of alternative methods |
|---|---|---|---|
| Step 1: Interpolation | Interpolation | Mean of spatio-temporally neighboring, non-mising points | Kriging, linear interpolation, nearest-neighbor interpolation kriging or more complex interpolation methods. |
| Step 2: Feature engineering | Feature engineering | Moving window averages, constant maps, space, time | Guided by domain knowledge or common statistical learning methods (e.g. greedy feature selection, polynomial features) |
| Step 3: Clustering | Classification | KMeans | Self-organising maps, Support Vector Machines, DB-SCAN or domain-guided classifications like Köppen climate classes (Köppen, 1884; Beck et al., 2018). |
| Step 4: Learning | Regression | Random Forest | Multiple Linear Regression, Neural Nets, Gradient Boosting, Gaussian Models. |

they are ill-equipped to incorporate the neighborhood relations with spatially extensive, gridded data. Spatial analogue searching algorithms such as the direct sampling approach by Mariethoz et al. (2012) and image inpainting (Kadow et al., 2020) explore multivariate spatial interpolation. Upscaling is a common, multivariate regression-based approach in Geosciences to gap-fill spatially incomplete observations but rely on at least one complete "donor-variable" or an additional, gap-free dataset to infer values of incomplete variables (Brocca et al., 2014; Kadow et al., 2020; Zhang et al., 2018; Zeng et al., 2015; Brajard et al., 2019; Ghiggi et al., 2019).

In summary, there exists a rich body of geoscientific literature on tailored solutions for individual gap-filling needs. However, no unified and modular solution exists that can be applied for any gap-filling scenario that might arise when working with Earth system observations. In the following, we introduce the multivariate gap-filling framework CLIMFILL that aims at overcoming the mentioned issues (Sect. 2). Section 3 describes a case study used for evaluating and benchmarking the framework. In Sect. 3.1 we describe the data that has been used to evaluate the skill of the framework. The choices used for benchmarking the framework in this study are outlined in Sect. 3.2. Finally, Sect. 4 discusses the results and provides a conclusion and an outlook on for possible future work.

## 2 CLIMFILL: A Generalised Framework for Infilling Missing Values in Multivariate spatio-temporal geoscientific data

We aim for a multivariate gap-filling framework that exploits the highly structured nature of Earth system observations to produce estimates for missing values. The framework builds upon previous research (van Buuren, 2018; Stekhoven and Bühlmann, 2012) and has enough flexibility to be tailored to fill missing values in a wide range of Earth observation datasets. To this end we aim at utilizing (1) spatial neighboorhood information, (2) temporal autocorrelation and (3) physical links between the



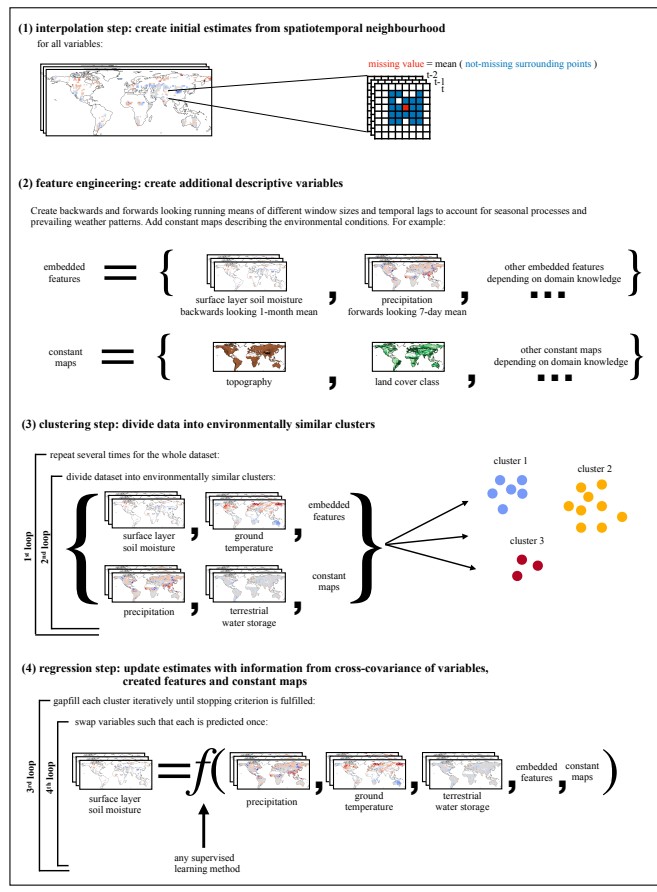

**Figure 2.** Overview on the structure of the gap-filling framework. The framework is divided into three steps. In the first step (Sect. 2.1), any missing value is gap-filled by an initial estimate from the spatio-temporal context. This step is called interpolation step. Here the spatio-temporal mean of observed values surrounding the missing value is used for each variable individually. In the second step (Sect. 2.2), embedded features are created to inform about time-dependent processes. In the third step, the data are divided into environmentally similar clusters (Sect. 2.3, Algorithm 1). In the forth step (Sect. 2.4, Algorithm 1), the inital estimates from step 1 are updated while accounting for the dependence structure among all considered variables. This is achieved by first grouping available data point into environmentally similar clusters and then iteratively updating the initial estimates using a supervised learning algorithm.

different variables expressed through their statistical dependence. With this design we aim at recovering both the marginal distributions and the dependence among variables at any location with missing values. The framework CLIMFILL (CLIMate data gap-FILL) works mutually, i.e. information available in each of the variables is used for filling the gaps of all the other variables. With this design we implicitly assume that if one variable is not observed at a certain space-time point, a subset of the other variables might be observed and can reconstruct the missing value while conserving the correlation structure among





all variables.

The framework is divided in four steps (Fig. 2): In a first step, initial estimates for all missing values are produced by spatio-temporal interpolation of each variable independently. In a second step, the data are pre-processed to enable the analysis of spatial and temporal dependence, which ultimately allows to uncover physical links among different variables. In the third step, the data are divided into environmentally similar clusters. In the forth step (learning step) the multivariate gap-filling happens: the initial estimates from the interpolation step are updated by an iterative procedure that aims to both reconstruct the dependence structure between the variables and to increase the accuracy of the initial estimates.

In the following, the newly developed iterative framework for gap-filling is described. CLIMFILL allows for a wide range of different options, creating a different instance of the framework for any missing value problem. A summary of the the necessary steps for setting up the framework, possible tweaks and extensions is given in Table 2. These include the process for coming up with initial estimates in step 1, feature engineering in step 2, as well as the selection of a clustering method in step 3 and the regression method used in step 4.

## 2.1 Step 1: Interpolation for integrating spatio-temporal context

The interpolation step creates initial estimates based on the spatio-temporal context of the gap. Interpolation methods that are typically used in geosciences, such as linear, bilinear or nearest neighbor interpolation as well as kriging can be used here (for examples see Table 2).

## 2.2 Step 2: Feature engineering informed by process knowledge

An important step in data driven modelling is taking care that the data consist of informative variables that represent the mechanisms at work. This creation of informative variables or "features" guided by expert knowledge is called feature engineering. For example, gap-free constant maps of describing properties of the land surface such as topography or land cover can be included. Furthermore, Earth observations often inform about time dependent processes like seasonal effects, weather persistence or soil moisture memory. To account for such antecedent and subsequent effects, backwards and forwards looking running means of different window size and temporal lags are considered. This is motivated by the Takens Theorem (Takens, 1981) and prior work on large-scale runoff estimation (Gudmundsson and Seneviratne, 2015). Given a variable $v_{i,j,t}$ at longitude $i$, latitude $j$ and time step $t$ we define the window size $s$ and time lag $l$ over which a running mean of a variable $v$ is computed:

$$v*_{i,j,t}(l,s) = \frac{1}{s}\left(v_{x,y,t-s-l} + v_{x,y,t-(s-1)-l} + ...v_{x,y,t-l}\right) \tag{1}$$





resulting in an embedded feature $v^*_{l,s}$ produced from variable $v$. The specific values for $s$ and $l$ can be informed by domain knowledge or identified through optimisation. For example, to account for the soil moisture memory effect, an embedded feature $v^*$ could be added that contains the average value of all soil moisture values at this point in a 3-month backwards

window ($s = 90$ days) from the current date ($l = 0$ days), corresponding to previous work indicating the soil moisture memory effect acts on "monthly to subseasonal time scales" (Nicolai-Shaw et al., 2016). For the application of data science methods, the data need to be rearranged in a table $\mathbf{X}$ build from all variables $v_1, ... v_n$ and derived features $v^*_1, ..., v^*_m$ as columns and space-time points as rows.

## 2.3 Step 3: Clustering the data into environmentally similar clusters

Depending on the climate regime and the season different physical processes might govern the local dependence among variables. Furthermore, geoscientific datasets are very large and the computational costs of supervised learning methods does often not scale linearly with the number of samples. We therefore split the data into $K$ environmentally similar clusters $\mathbf{X}^{(1)}, ..., \mathbf{X}^{(K)}$ (Algorithm 1, line 5) in which the multivariate gap-filling happens (Algorithm 1, second loop, line 6+17). This grouping is done in such way that grid points can be in different clusters depending on the time step. For example, a grid point in the Mediter-

ranean area can be in a different cluster in winter than in summer, accounting for seasonally varying climate phenomena such as changing soil moisture regimes (Seneviratne et al., 2010). All data are transformed to have zero mean and standard deviation of one.

In each of the clusters, the initial estimate of the missing values is further refined using an iterative procedure. For stabilising

the results and to reduce the risk of discontinuities at the cluster edges, the clustering procedure is repeated $E$ times with different numbers of terminal clusters on copies of the data $\mathbf{X}^{(1)}, ..., \mathbf{X}^{(E)}$. We call these $E$ different clustering results "epochs". In the end, the estimates from the $E$ different clusterings are averaged for the final result (Algorithm 1, outer first loop, line 3-5,19-20).

## 2.4 Step 4: Optimising the initial estimates by accounting for the dependence between variables

In the fourth step, the initial estimates are updated by accounting for the dependence between variables. Within each of the clusters in epoch $e$ and cluster $k$, $X^{(e,k)}$, the algorithm repeatedly iterates over the variables until convergence is reached. This procedure builds upon the MissForest algorithm by Stekhoven and Bühlmann (2012). For each variable $v$, a supervised learning model is fitted to the cluster to predict originally missing values in all variables based on the remaining features. This core mechanism of CLIMFILL is detailed in the inner, forth loop of Algorithm 1 (line 8 to 15): The current variable is

selected from the cluster as predictand $\mathbf{y}^{(e,k)}_v$. All other columns of $\mathbf{X}^{(e,k)}$ form the predictor table $\mathbf{X}^{(e,k)}_{-v}$, where $-v$ denotes the set of all variables and features except $v$. Subsequently both $\mathbf{y}^{(e,k)}_v$ and $\mathbf{X}^{(e,k)}_{-v}$ are divided into two sets of data points: (1) all data points where $\mathbf{y}^{(e,k)}_v$ was originally observed are used to fit the supervised learning method $\mathbf{y}^{(e,k)}_{v,o} = f(\mathbf{X}^{(e,k)}_{-v,o})$ and (2) all data points where $\mathbf{y}^{(e,k)}_v$ was missing $\mathbf{y}^{(e,k)}_{v,m}$ are predicted from the fitted function and overwrite the former estimates: $\mathbf{y}^{(e,k),updated}_{v,m} = f(\mathbf{X}^{(e,k)}_{-v,m})$. Note that the training data most likely include originally missing values in the predictor variables.





Here, the estimates from the interpolation step play the role of giving an initial estimate for the first loop of the iterative procedure. Once the algorithm has iterated over all the variables, each missing value has been updated once (Algorithm 1, third loop, line 7+16). The algorithm is stopped (stopping criterion) once the change in the estimates for the missing values is small between iterations (convergence) or a maximum number of iterations is reached (early stopping).

Note that the framework is set up such that each cluster applies the same supervised learning method but learns different weights. The hyperparameters for the supervised learner can differ for each variable and can be optimised e.g. through cross validation. With these choices the model is flexible to tailor its hyper-parameters individually to each variable and the regression weights individually to each cluster.

---

**Algorithm 1** Pseudo-code algorithm of the CLIMFILL clustering and learning step (step 3 and 4), where $E$ is the number of epochs, $K$ is the number of clusters, $n$ is the number of variables and $m$ the number of features. $\mathbf{X}_{-v}$ refers to the data table with all variables except $v$. Algorithm and pseudo-code are adapted from Stekhoven and Bühlmann (2012).

---

1: $\mathbf{X}$ is a matrix containing all variables and features as $n + m$ columns and all data points as rows.

2: Create a mask of missing values $\mathbf{M}$ in the same shape as $\mathbf{X}$, where $\mathbf{M}$ is **true** where $\mathbf{X}$ is missing and **false** where $\mathbf{X}$ is observed. Note that missing values are only present in variables, not in features.

3: **for** epoch $e = 1, 2, \ldots, E$ **do**

4:    Copy $\mathbf{X}$ to $\mathbf{X}^{(e)}$

5:    Randomly select number of clusters for this epoch $K^{(e)}$

6:    Split $\mathbf{X}^{(e)}$ into $K^{(e)}$ clusters $\mathbf{X}^{(e,k)}$ using an unsupervised classification method.

7:    **for** cluster $k = 1, 2, \ldots, K^{(e)}$ **do**

8:       **while** stopping criterion not reached **do**

9:          **for** variable $v = 1, 2, \ldots, n$ **do**

10:             Define current variable as predictand $\mathbf{y}_v^{(e,k)}$ and all other columns of $\mathbf{X}^{(e,k)}$ as predictors $\mathbf{X}_{-v}^{(e,k)}$.

11:             Define $\mathbf{y}_{v,o}^{(e,k)}$ as all data points in $\mathbf{y}_v^{(e,k)}$ where $\mathbf{M}$ is **false**, and $\mathbf{y}_{v,m}^{(e,k)}$ as all data points where $\mathbf{M}$ is **true**.

12:             Define $\mathbf{X}_{-v,o}^{(e,k)}$ as all data points in $\mathbf{y}_v^{(e,k)}$ where $\mathbf{M}$ is **false**, and $\mathbf{X}_{-v,m}^{(e,k)}$ as all data points where $\mathbf{M}$ is **true**.

13:             Fit the regression model $\mathbf{y}_{v,o}^{(e,k)} = f(\mathbf{X}_{-v,o}^{(e,k)})$ where $f$ denotes any supervised learning method.

14:             Create an updated estimate with the fitted regression model $\mathbf{y}_{v,m}^{(e,k),updated} = f(\mathbf{X}_{-v,m}^{(e,k)})$.

15:             Replace $\mathbf{y}_{v,m}^{(e,k)}$ with the new updated $\mathbf{y}_{v,m}^{(e,k),updated}$ in $\mathbf{X}^{(e,k)}$.

16:             Update stopping criterion.

17:          **end for**

18:       **end while**

19:    **end for**

20:    Combine all $\mathbf{X}^{(e,k)}$ back to $\mathbf{X}^{(e)}$.

21: **end for**

22: Calculate mean over all epochs $\mathbf{X} = \frac{1}{E} \sum \mathbf{X}^{(e)}$ and save final result.

---





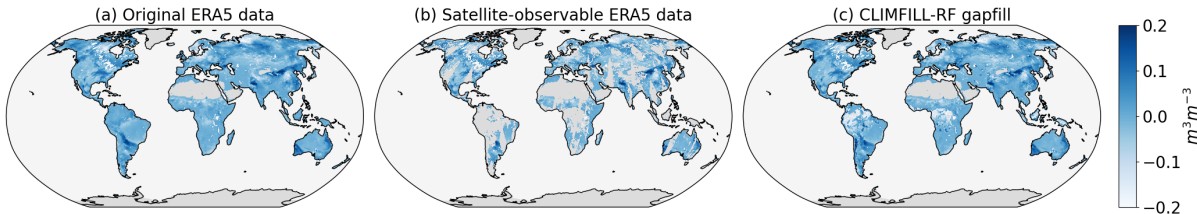

**Figure 3.** Comparison of (a) the original naturally gap-free ERA5 reanalysis, (b) the same data but only satellite-observable values are shown, and (c) the gap-fill created from CLIMFILL-RF after starting with the gappy data in (b) in example snapshot of ERA5 surface layer soil moisture anomaly on 1 August 2003. CLIMFILL-RF successfully reconstructs major anomalies in surface layer soil moisture for this day. The anomalies are calculated by substracting the 10-year mean of 2003-2012.

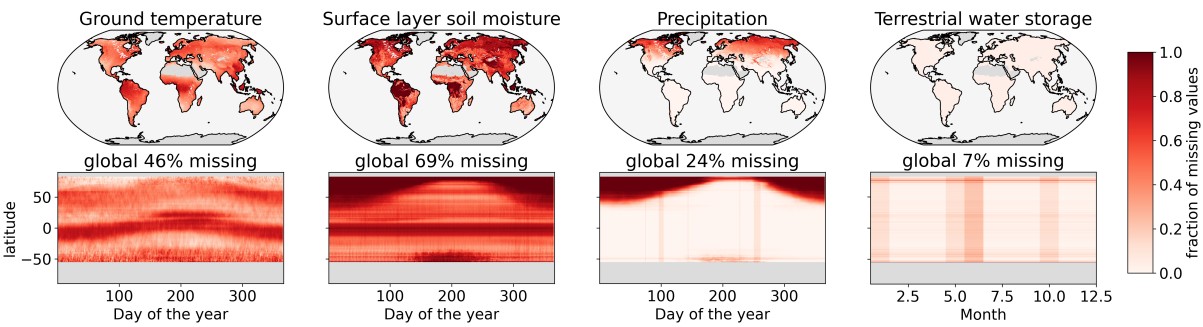

**Figure 4.** Fraction of missing data in ground temperature from MODIS, ESA-CCI soil moisture, GPM precipitation and GRACE terrestrial water storage observations in the years 2003-2012. Upper panels show fraction of missing data per land points on the ERA5 grid, lower panels show fraction of missing values per latitude and day of the year. The data are down-sampled to daily values, except GRACE which has monthly resolution.

## 3 Testing and Benchmarking the CLIMFILL-Algorithm

### 3.1 Data

Since the original values that need to be gap-filled are unobserved, we fall back on naturally gap-free atmospheric reanalysis data for benchmarking the framework. We use 10 years (2003-2012) of land-only global reanalysis data from ERA5 at 0.25 degree resolution (see Hersbach et al. 2020). ERA5 is chosen as a gap-free dataset for the "perfect dataset approach" because of its advanced representation of land surface processes (Hersbach et al., 2020) and improved agreement of relevant surface variables with available observations (Martens et al., 2020; Tarek et al., 2020; Albergel et al., 2018). The missingness patterns of satellite observations in the same period are extracted, regridded to ERA5 resolution and applied to the corresponding ERA5 variable. In other words, only the part of the ERA5 data that would have been observable by satellite are retained. In this "perfect dataset approach", the "true" values of the variables at the locations of the missing values are known and can be compared





with the estimates of the gap-filling framework (see Figure 3). This analysis is constrained to orbiting satellite remote sensing
datasets and excludes in situ observations and gridded observations for the purpose of developing the framework. We note
however that the framework is naturally extendable to include more satellite observations, and in situ observations that can be
treated as a very sparse gridded product.

The hourly ERA5 data are aggregated to daily resolution. The aggregation function for each variable is chosen to be con-
sistent with the satellite products (e.g. daily sums for precipitation and daily average for soil moisture, see Supplementary
Table A1). Since GRACE is only available in monthly resolution, we up-sample the data by linearly interpolating the monthly
values to daily resolution. Permanently glaciated areas and deserts are masked. We extract the missingness pattern from four
satellite remote sensing datasets related to land climate interactions and apply it to the ERA5 dataset: ESA-CCI surface layer
soil moisture (Gruber and Scanlon, 2019; Dorigo et al., 2017; Gruber et al., 2017), MODIS ground temperature (Wan et al.,
2015), GPM precipitation (Huffmann et al., 2019) and GRACE terrestrial water storage (Swenson, 2012; Landerer and Swen-
son, 2012; Swenson and Wahr, 2006) on daily timescale. These variables represent central interactions between soil moisture
and climate that drive land water and energy balance through the soil moisture-temperature and the soil moisture-precipitation
feedbacks (Seneviratne et al., 2010). Selecting both microwave remote sensing measures of surface layer soil moisture and
total water storage of the land surface is a compromise aiming at including as much possible information of root zone soil
moisture as there is available via remote sensing.

There are ubiquitous missing values in the selected satellite observations (Figure 4). Since the missingness patterns are only
partially overlapping the selected set of variables is a good candidate for mutual gap-filling. Ground temperature is missing
where there is cloud cover, with the maximum of missing values in the inner tropics and extratropical strom tracks, moving
along latitudinal bands throughout the year. Almost half of the values globally (46%) of ground temperature are missing in
the ten considered years. Surface layer soil moisture is only observed in 31% of all cases. It is missing where there is ice
or snow cover or when vegetation is too dense. This is the most complicated missingness case, because of the high fraction
of missing values and the considerable amount of land mass where high vegetation cover prevents retrieval at all times. For
precipitation, a quarter of the values are missing (24%), and only in high latitudes during winter. In the GPM remote sensing
precipitation dataset values in the presence of surface snow or ice are masked because of poor sensor quality (Huffmann et al.,
2019). In postprocessing, Huffmann et al. (2019) use a sophisticated kalman-smoother time interpolation to fill the gaps from
the retrieval. From available metadata, we retrieved the originally missing maps to be able to quantify the added value of mutual
gap-filling for precipitation. Terrestrial water storage is only missing if the global measurement is discarded due to instrument
failure or during calibration missions (Landerer, 2021), leading to individual time slabs missing, and only 7% missing values.

## 3.2 CLIMFILL-RF: Settings of the CLIMFILL framework used for benchmarking

The CLIMFILL framework allows for a wide range of individual settings to tailor it to the specific gap-filling use case. In
each of the four steps, a method needs to be chosen to perform the specific task of this step. There is a large pool of methods





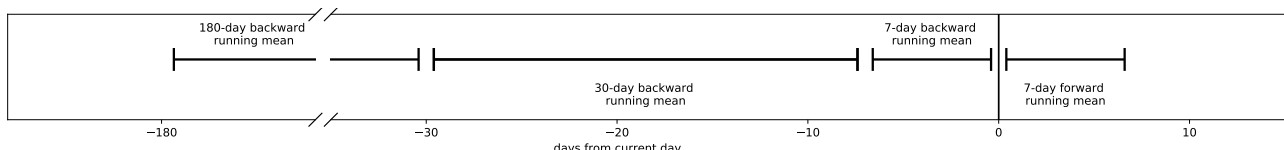

**Figure 5.** Time lags and window sizes of embedded features used in this study.

that can be used, for examples see Table 2. In the following, we describe the settings of the framework that are used for this benchmarking experiment and call this particular instance CLIMFILL-RF, denoting the Random Forest method used at the

core of the algorithm.

For the first step (interpolation) initial estimates are generated through simple interpolation by applying a 3d running mean for each variable independently. If a data point of a variable, $v_{i,j,t}$, is missing, the initial estimate is calculated by the mean of its non-missing surrounding points in space and time. Here we consider a 5-pixel side length, corresponding to a distance of

1.25 degree in space and 5 days in time. If a point cannot be filled because all the values in the neighbourhood are missing as well, the points is filled by the local monthly climatology. Any remaining missing points are filled by the local temporal mean, or, if not available, the global mean of the variable.

In the second step (feature engineering), we create embedded features of 7-day ($s = 7, l = 0$), 1-month ($s = 23, l = 7$) and

6-month backward ($s = 150, l = 30$) and a 7-day forward ($s = 7, l = -7$) running means in such way that the windows are not overlapping (see Eq. 2.2 and Fig. 5). This way 3 additional features are created for each variable. Constant maps of altitude, topographic complexity, land cover class and land cover height from ERA5 as well as latitude, longitude and time are added to the list of features and copied for each time step. Furthermore, precipitation is divided into a log-scaled precipitation-amount variable and a binary precipitation-event variable to treat its inherent non-normality. The above proceedure thus results in a set

of 34 features: The four variables, where precipitation is divided into two features, the four embedded features of each of the five variables, totalling in 20 embedded features, the six constant maps and latitude, longitude and time information.

In the third step (clustering step), the data are divided into clusters. Here a k-means algorithm is considered and the data are partitioned three times with different number of clusters, where the number of clusters is randomly drawn between 50 and 150.

These limits are chosen such that the number of data points per cluster is sufficiently large to ensure that the regression models can be calibrated efficiently, but not too small such that no individual clusters consist of missing values entirely.

In the fourth step (learning step), we use a Random Forest regressor as supervised learning function. Random Forests have have favorable properties for gap-filling applications: they can handle mixed types of data, are scale-able to large amounts of data and non-parametric, i.e. adaptive to linear and non-linear relationships (Tang and Ishwaran, 2017). The hyper-parameters

of the supervised learning functions are determined via leave-one-out cross-validation on clustered ERA5 data between 2015 and 2020 downscaled to 2.5 degrees resolution, where one fold is one year. The cross-validation optimises the number of trees,



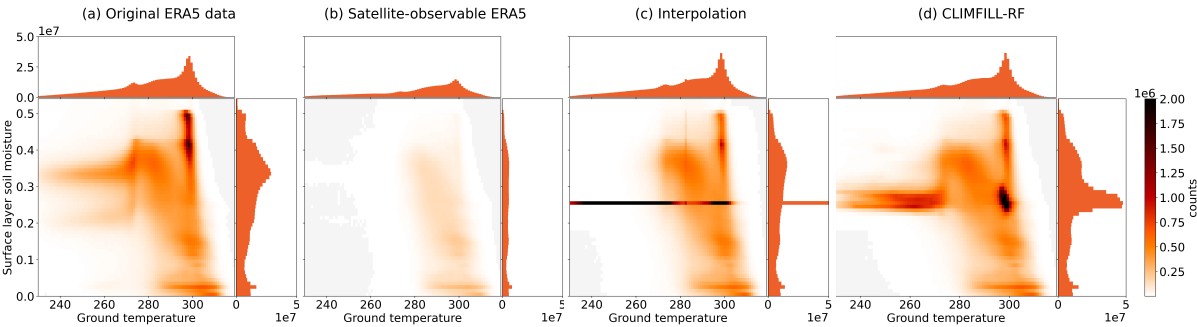

**Figure 6.** Bivariate and univariate histograms of surface layer soil moisture and ground temperature in (from left to right) original ERA5 data, the subset of the original ERA5 data that would have been observable by satellite, gap-filled throught univariate interpolation and with CLIMFILL-RF gap-filling. For bivariate distributions of other variable pairs see Supplementary Figure A1

the minimum number of samples for a leaf node, the maximum number of features to be considered for each split and whether to use bootstrap samples for tree building.

## 3.3   Benchmarking against univariate interpolation

The objective of the CLIMFILL framework is to not only reconstruct variables separately, but also to recover multivariate dependencies. In the first part of the results, we illustrate the improvement of the multivariate gap-filling framework CLIMFILL-RF compared to the univariate, spatiotemporal interpolation that takes place in the first step of the framework.

Figure 6 shows the bivariate distribution of surface layer soil moisture and ground temperature globally for the whole time
period (all other possible combinations of bivariate distributions are shown in Supplementary figure A1). Only looking at the part of the data that is observable from space (Figure 6 b) misses larger chunks of the original bivariate distribution. Results after interpolation show a collapsed distribution, where large areas have identical soil moisture values. This is indicating the areas where spatio-temporal interpolation failed because no close measured value could be found and the mean was inserted instead (see Sect. 2.1). CLIMFILL-RF recovers the shape of the original distribution and is able to overwrite unrealistic surface layer
soil moisture values. Thus it generally provides an improved estimate of the bivariate distribution of surface layer soil moisture and ground temperature such that it is closest to the original ERA5 data in spite of knowing only satellite-observable points.

While Fig. 6 and Supplementary Fig. A1 enable a visual inspection of selected variable pairs, they do not allow for a quantitative assessment of the similarity of the multivariate distributions of observed and simulated variables. To overcome
this issue we apply a scalar measure of multivariate similarity. This measure compares the multivariate distance between two datasets or distributions, where a value of zero means that the two samples are from the same distribution, and a positive value indicates the relative distance between two distributions. In this study, we use the Bhattacharyya distance (Bhattacharyya,



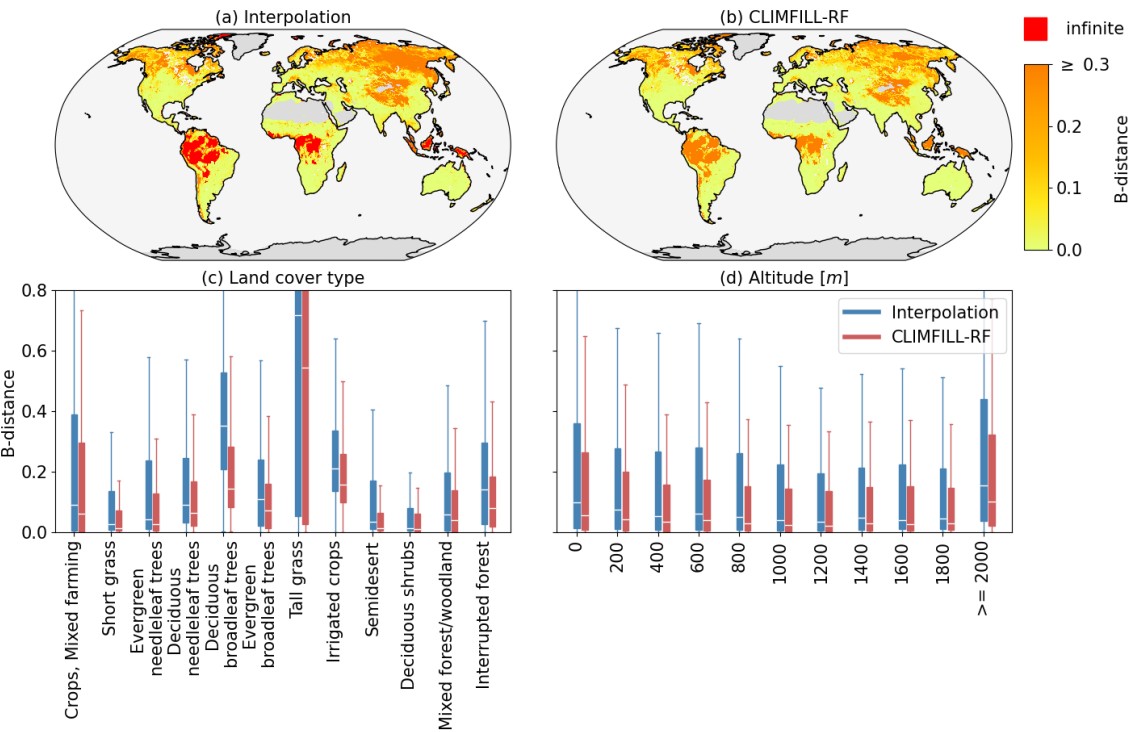

**Figure 7.** Multivariate B-distance for interpolation and CLIMFILL-RF gapfill. Map of B-distance of univariate interpolation (a) and CLIMFILL-RF (b) as well as B-distance per land cover type (c) and altitude (d) for interpolation gap-fill and CLIMPUTE-RF gap-fill in real missingness case. Land cover type and altitude are extracted from ERA5. Boxplots show the median as white line, the box as the quartiles and the whiskers at 1.5 times of the quartile length over all landpoints with the specified land cover type or altitude, respectively. Infinite values in the boxplots are replaced with the maximum, not-infinite value.

1943) (B-distance). The B-distance is a general measure to quantify the distance of two multivariate distributions, taking into account both the similarity in mean and covariance of both distributions. For samples of two multivariate normal distribution

with means $\mu_1$, $\mu_2$ and covariances $\Sigma_1$, $\Sigma_2$, the Bhattacharyya distance is defined as

$$B - distance = \frac{1}{8}(\mu_1 - \mu2)^T \Sigma^{-1}(\mu_1 - \mu_2) + \frac{1}{2}\ln\left(\frac{\det\Sigma}{\sqrt{\det\Sigma_1 \det\Sigma_2}}\right) \qquad (2)$$

where $\Sigma$ is the mean of $\Sigma_1$ and $\Sigma_2$. The first term is a measure of similarity of the mean between two samples, and the second term is a measure of similarity of their covariances. Although the data considered may not be normally distributed we rely here on the normal approximation of the B-distance to facilitate a quantitative comparison of the considered gap filling methods at

a reasonable computational cost.





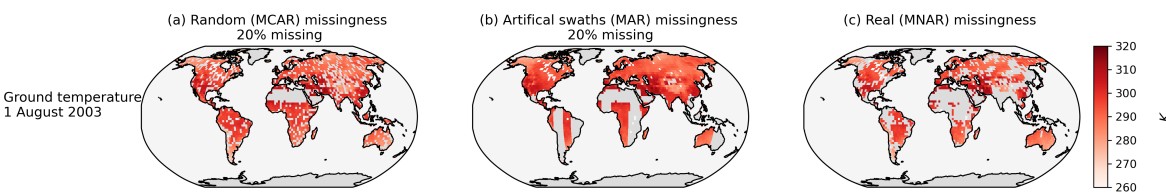

**Figure 8.** Comparison of (a) artificial (a) random and (b) swaths-only missingness in example snapshot of ERA5 ground temperature on 1st of August 2003 with. Random missingness was created by randomly sampling without replacement from the pool of all gridpoints on land at all timesteps in the desired fraction of missing values. In swaths-only missingness we create Note that the two missingness patterns are not exactly the same for each day and variable to allow for mutual learning.

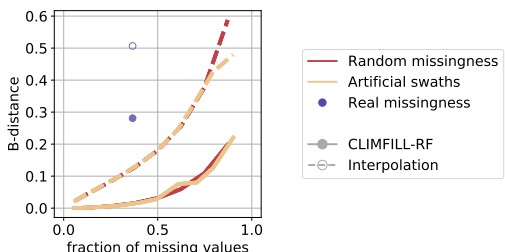

**Figure 9.** Median performance of gap-filling with CLIMFILL-RF on different missingness patterns and fractions of missingness expressed in B-distance (for more detail see text) per variable. Gap-filling for random missingness and artifical swaths is executed for a range of fraction of missing values and denoted as a line, while real missingness is only one case depicted as point. The metrics are calculated over each timestep for all not satellite-observable values of gridpoints on land and the median of all landpoints is plotted.

Overall, the B-distance is lower for CLIMFILL-RF than for interpolation globally (Fig. 7). The largest improvement is in temperate and boreal regions, where a high fraction of missing values inhibits the performance of interpolation. In parts of the inner tropics the B-distance of the interpolation gap-fill is not defined (in Fig. 7 indicated with dark grey color). Here the gap-fill estimate from interpolation is the same in every time step, because the high vegetation cover causes the satellite to never observe surface layer soil moisture in this area. Leads to an all-zero covariance and therefore theoretically infinite, practically undefined B-distance. These points have been removed in Fig. 7 (c) and (d) to improve readability. Taking a closer look at the results by dividing the global map into types of vegetation and altitudes shows that the B-distance improves from interpolation to CLIMFILL-RF for all altitudes and almost all land cover types. This indicates an improvement of multivariate features in CLIMFILL-RF gap-fill globally for a wide range of environmental conditions. Overall CLIMFILL-RF has a higher skill in reconstructing the multivariate dependence structure of the original ERA5 data compared to univariate interpolation.



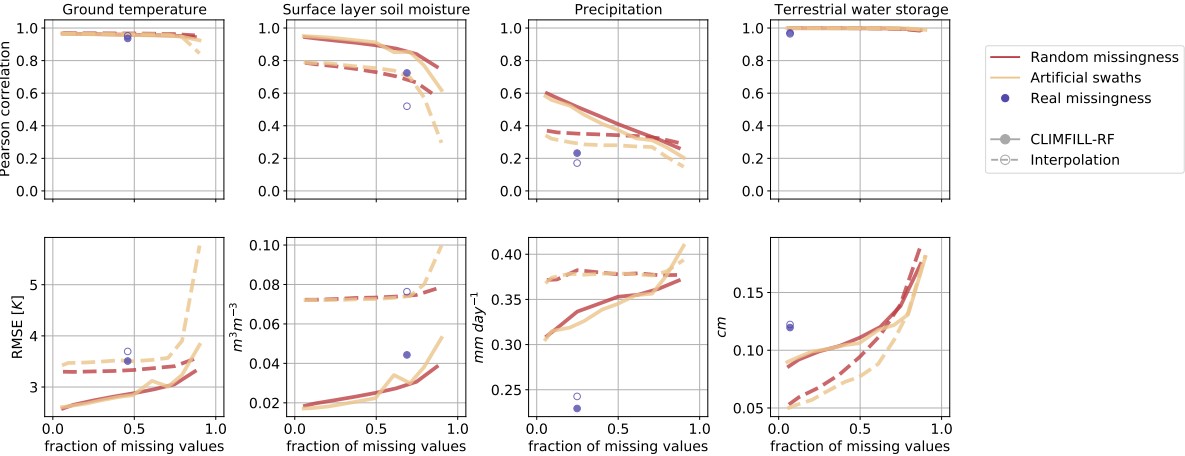

**Figure 10.** Median performance of gap-filling with CLIMFILL-RF on different missingness patterns and fractions of missingness expressed in three metrics: pearson correlation, root mean square error (RMSE) and B-distance (for more detail see text) per variable. Gap-filling for random missingness and artifical swaths is executed for a range of fraction of missing values and denoted as a line, while real missingness is only one case depicted as point. The metrics are calculated over each timestep for all not satellite-observable values of gridpoints on land and the median of all landpoints is plotted.

## 3.4 Data-constrained upper perfomance limits

Missing values in Earth observation data are often present in a large proportion and a complex missing-not-at-random pattern. These characteristic properties of gappy Earth observation data can inhibit gap-filling. We therefore are interested in carving

out the envelope of data properties in which gap-filling can be successful and see the deterioration of performance with increasing data sparsity and increasingly complex missing value patterns. Using the four considered ERA5 variables we test the framework in idealised, simpler missingness patterns. In these additional experiments, we delete data according to a (1) MCAR random missingness pattern and (2) imitating satellite swaths, effectively creating MAR missingness patterns (Fig. 8). Both patterns are applied for fractions of missing values between 5% to 80% for each of the variables. We performed these

experiment on a downscaled 2.5 degrees resolution ERA5 data because of computational constraints.

Multivariate B-distance (Figure 9) and univariate statistical performance measures (Figure 10) for all performed experiments comparing original and gap-filled values. With increasing fraction of missing values, the two artificial missingness cases increase in error, increase the B-distance and decrease in correlation. Once more than 80% of the values are missing, the

gap-filling breaks down because not enough observed values are available for the iterative procedure to converge to a meaningful result. Random and artificial swath missingness show similar deterioration with increasing fraction of missing values, but values missing completely at random tend to be easier to estimate at all fractions of missing values. Gap-filling random missingness is the easiest case, since it is likely that neighboring or environmentally similar points are observed. MAR missingness



exposes large patches of missing values, therefore making spatiotemporal interpolation less effective and therefore decreasing

the gap-filling performance as compared to MCAR. Since the MNAR missingness case is the most complex missingness pattern, these additional experiments serve as upper limits of the performance in the real case.

When moving from the artificial patterns of missingness to the real case (dots and circles in Fig. 10), the deterioration in performance is different for each of the variables. For ground temperature, a spatially and temporally smooth variable, the

interpolation is already quite a good first guess, which is only slightly improved in CLIMFILL-RF. In this case study, we found the biggest improvement compared to interpolation for surface layer soil moisture despite its large fraction of missing values. This could be due to the fact that surface layer soil moisture exposes missingness in areas where other variables are observed, for example in the tropical forests, such that learning in this area is easier. Additionally, variable selection is centered around soil moisture, and soil moisture is a key variable of land hydrological processes. The most difficult case is precipitation. Despite

the additional pre-processing step to account for its non-normality, the low precision precipitation estimates were only slightly improved with CLIMFILL-RF and it is difficult to improve the result of the initial interpolation. Precipitation is influenced by a lot of processes that are not captured within the four selected variables. For example, frontal rain patterns are mostly not explained by land surface properties but are governed by large scale circulation. This is a challenging case and could still furthermore be improved, for example by adding wind patterns to capture more synoptic features. Terrestrial water storage

contains only a small fraction of missing values (7%), but its gap-filling could be hampered by its monthly resolution that does not co-vary enough with the other variables. Introducing an additional bias correction step could help alleviate these problems.

### 3.5 Recovery of regional and local land-climate dynamics

For any gap-filling framework to be useful for both scientific and practical applications it needs to be able to recover essential properties of the phenomena of interest. The coupling of energy and water between land and atmosphere at the land surface

is a central, multivariate property of land climate interactions that is currently underestimated in satellite data (Hirschi, 2014). By comparing CLIMFILL-RF gap-fill with the subset of data that are observable by space, i.e. the gappy ERA5 data (Fig. 3) we explore the role of missing values in this problem. In particular we show that leaving gaps in satellite data unfilled leads to biases in estimates of regional and local climate feedbacks and how the CLIMFILL framework contributes to overcoming this issue.


Figure 11 showcases the mean seasonal cycle of the variables for selected IPCC reference regions (AR6 regions, see Iturbide et al. 2020). Surface layer soil moisture, ground temperature and precipitation suffer from gaps in the winter months in mid to high latitude regions like Western & Central Europe and South-West North America. In tropical regions like Central Africa and South-East Asia, especially soil moisture estimates suffer from little data availability. The missing values result in

a noisy signal and biased values in regional estimates from the satellite-observable data. CLIMFILL-RF alleviates the noise and reduces the bias for surface layer soil moisture and ground temperature for these regions with low satellite coverage better than the interpolation estimates. The largest relative difference is in the surface layer soil moisture estimates. For surface layer

**Figure 11.** Mean seasonal cycle over selected IPCC reference regions (AR6 regions, Iturbide et al. 2020) in original ERA5 data, satellite-observed ERA5 data and gap-filled CLIMFILL-RF data. The selected regions are in areas with the largest fractions of missing values globally and show exemplary advantages and problems of the framework, see text. For all other AR6 regions see Supplementary Fig. A2

soil moisture and ground temperature especially the amplitude of the signal is reconstructed, but also the bias is reduced in all regions (see Supplementary Fig. A2). Precipitation and terrestrial water storage estimates show little change.


Soil moisture-temperature coupling plays an important role for the development of heat extremes (Seneviratne et al., 2010; Vogel et al., 2017; Wehrli et al., 2019). This feedback can be described by the correlation between the soil moisture anomaly $sm_{anom}$ and the number of hot days (NHD). The correlation can expose "hot spots" of soil moisture-temperature coupling where hot extremes can be exacerbated (Mueller and Seneviratne, 2012; Hirschi, 2014) and is central for representing com-



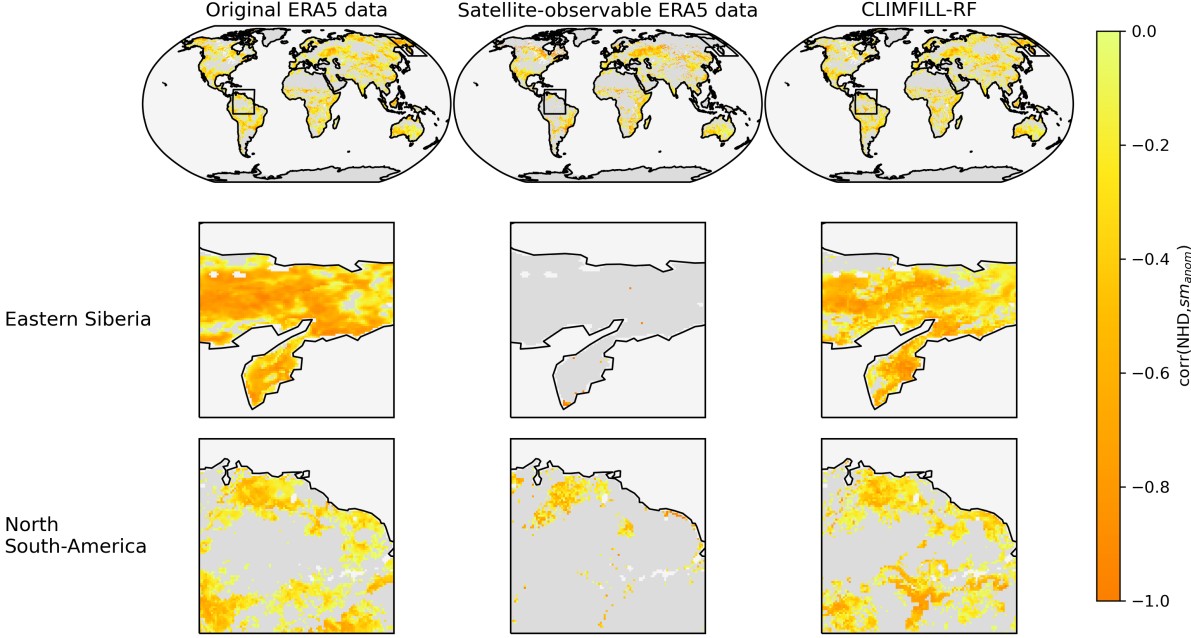

**Figure 12.** Correlation between number of hot days (NHD) and soil moisture anomaly $sm_{anom}$ in the selected time period for original ERA5 data, satellite-observable ERA5 data and CLIMFILL-RF gap-fill. The selected regions are in areas with the largest fractions of missing values globally and show exemplary advantages of the framework, see text. Methodology from Mueller and Seneviratne (2012) and Hirschi (2014).

pound extreme events at the land surface, such as droughts and heat waves. We compute this correlation for original ERA5 data, satellite-observable ERA5-data and the CLIMFILL-RF gap-fill. Hirschi (2014) note that the coupling strength between remotely sensed soil moisture and NHD is qualitatively similar, but underestimated in satellite observations compared to a precipitation-based soil moisture estimate from interpolated weather station data (CRU dataset, Harris et al. (2020b)). A similar effect can be found in ERA5 data when only satellite-observable data points are taken into account. Comparing Fig. 12

shows that removing data from ERA5 that would not have been observable via space leads to a deterioration of soil moisture-temperature coupling strength, especially in the showcased regions that have sparse surface layer soil moisture observations such as tropical forests and high latitudes. In these areas, CLIMFILL-RF is able to alleviate the underestimated coupling (Fig. 12) and successfully reconstructs the correlation between NHD and $sm_{anom}$ in these regions. This is highlighting that missing values in Earth observation can bias process analysis and multivariate gap-filling can help alleviating these biases and recover

important dynamics and dependencies between variables which would have been dampened or lost in gappy satellite-data alone.





# 4 Discussion and conclusions

Gaps in Earth observations are unavoidable and lead to a fragmented record of observational data. CLIMFILL is a framework for gap-filling multivariate gridded Earth observations that estimates missing values by taking into account the spatial, temporal and the multivariate context of a missing value. In doing that CLIMFILL mines the highly structured nature of geoscientific datasets and bridges the gap between interpolation-centered approaches common to geosciences and multivariate gap-filling methods from statistical literature. In contrast to popular up-scaling approaches, CLIMFILL does not need a gap-free gridded "donor" variable for learning and can digest any gap structure in the provided data, including spatial gaps, temporal gaps and non-overlapping observations from different datasets. Furthermore, by clustering the global data into environmentally similar points, we tailor the multivariate gap-filling to the needs of datasets spanning global, highly diverse ecosystems and changing land-atmosphere interactions. This approach also decreases computing time such that high resolution gap-filling is possible (not shown). The highly flexible nature of CLIMFILL does not imply a physical model, but allows important physical dependencies to be imprinted in the dataset before gap-filling through feature engineering. This way, CLIMFILL can be tailored to many geoscientific use cases. In summary, CLIMFILL can successfully fill gaps in fragmented Earth Observation datasets, while maintaining the physical dependence structure among the considered variables. To this end, the CLIMFILL framework contributes to decreasing the inherent fragmentation of earth observations and enables usage of multiple gappy satellite observations simultaneously.

We have tested and bench-marked CLIMFILL in an exemplary setting of land hydrology reanalysis data. To this end this data have been deleted to match missing values in satellite observations in a "perfect dataset approach". This case study shows that seeing only satellite-observable data without filling the gaps creates biased, noisy regional estimates and destroys the dependence structure in multivariate settings. CLIMFILL is able to recover this dependence structure in land-atmosphere coupling and hence enables process investigation in gappy, multivariate observations. Quantified with the multivariate B-distance we show that this recovery improves the dependence structure globally across almost all land covers and altitudes compared to interpolation. The largest improvements are in temperate and boreal regions, although these are areas with large patches of low numbers of observed points. In summary, CLIMFILL is able to recover the dependence structure among several variables, contrasting results obtained when missing values are not gap-filled or treated without considering multivariate aspects. Thereby CLIMFILL enables a physically consistent interpolation of the resulting gap free dataset.

Interestingly, the case study showed that the benefit of CLIMFILL compared to interpolation is not equally large across variables. The selected group of variables and their individual missing value patterns are central for the success of multivariate gap-filling. Learning from the other variables is highly beneficial in gap-filling surface layer soil moisture estimates, although it has the largest fraction of missing values. Since the framework is targeted at recovering the the physical dependence structure across variables, the improvement in univariate measures like correlation and bias tend to be improved at a smaller scale than the multivariate dependence structure. The case study also highlights that information from other available variables can



indeed be beneficial for gap-filling if process knowledge is used when selecting a sub-set of variables and suggests the potential power of the framework if even more dependent and important variables are included in the multivariate gap-filling process.

475 Although the selected observations are small in number (only four variables considered), high in their respective fraction of missing values and complex in their pattern of missing values (always missing not at random), the multivariate gap-filling with CLIMFILL successfully improves estimates compared to univariate interpolation. This is likely related to the high correlation among the variables, which can to some degree counteract the complex missingness. Idealised experiments with simpler missingness patterns and different fractions of missing values within these four variables show that CLIMFILL improves upon univariate interpolation in all cases for all considered metrics, but that multivariate gap-filling is easier with smaller fractions 480 of missing values and less complex missingness patterns. The high correlation and low error scores for low fractions of missing values indicate that the four included variables represent important processes and are explanatory for each other, i.e. their mutual dependence is expressive enough to conduct meaningful gap-filling.

In conclusion, we have presented a multivariate gap-filling framework that uses spatial, temporal and multivariate information to create estimates for missing values. This framework has been successfully applied in a case study centered around land 485 hydrology remote sensing observations. The modularity and flexibility of the proposed gap-filling framework make it applicable to all kinds of Earth observation data once suitable settings are chosen by applying knowledge of the important physical processes represented in the data. CLIMFILL can be used for multivariate, observation-only process analysis or help including relevant but gappy observations into data assimilation or reanalysis. in situ data could possibly be included as well if treated 490 as a very sparsely gridded data where the area of representation for the point measurement is accessed (see e.g. Nicolai-Shaw et al. 2015). A natural next step could be to apply this gap-filling mechanism on a larger number of relevant observed variables and create a consistent, gap-free reconstruction of land hydrology.

Missing values in Earth observations will remain unavoidable. However, the intrinsic motivation should be to reduce gaps 495 in observations. Enhancing sensors, developing new measurement techniques or closing gaps in observational networks are three possible directions of innovation that could help reduce missing information. This endeavour however must start with an assessment of the information completeness of existing observations, for example by applying methods from information theory. It should aim at closing the largest gaps first, for example in terms of available variables, sampled ecosystems or in physical space. Reducing the complexity of missing information in Earth observations can be a large step towards better 500 observational estimates of crucial Earth system processes.

*Code and data availability.* The current version of CLIMFILL is available from the project website: https://github.com/climachine/climfill under the Apache 2.0 License. The exact version of the model used to produce the results used in this paper is archived on Zenodo (http://doi.org/10.5281/zenodo.4773664), as are scripts to run the model and produce the plots for all the simulations presented in this paper.





CLIMFILL was written in python (Python Software Foundation, https://www.python.org/) with core packages including xarray (Hoyer et al.,
505   2020), numpy (Harris et al., 2020a), matplotlib (Hunter, 2007), scikit-learn (Pedregosa et al., 2011), regionmask (Hauser, 2021) and scipy
(Virtanen et al., 2020). The used ERA5 data are publicly available at: https://www.ecmwf.int/en/forecasts/datasets/reanalysis-datasets/era5
(last accessed: 16th February 2021).



**Figure A1.** Improvement of multivariate distribution with CLIMFILL gap-filling: 2D-histogram of all combinations of variables for not satellite-observable values in original ERA5 data, interpolation and CLIMFILL-RF.



**Figure A2.** Mean seasonal cycle over all IPCC reference regions on land (AR6 regions, as described in Iturbide et al. 2020) in original ERA5 data, satellite-observed ERA5 data and data gap-filled with CLIMFILL-RF.



**Figure A3.** Supplementary Figure A2 continued





**Figure A4.** Supplementary Figure A2 continued





**Table A1.** Mapping of ERA5 variables with satellite observations.

| satellite observation | ERA5 variable | daily aggregation | unit |
|---|---|---|---|
| ESA-CCI surface layer soil moisture | volumetric soil water layer 1 $swvl1$ | daily mean | $m^3 m^{-3}$ |
| MODIS ground temperature | ground temperature $skt$ | daily mean | $K$ |
| GPM precipitation | total precipitation $tp$ | daily sum | $mm\ day^{-1}$ |
| GRACE terrestrial water storage | volumentric soil water layer 1 to 4, snow depth $sd$ and lake cover $cl$ multiplied with lake depth $dl$ | anomalies of daily sums compared to GRACE baseline (2004-2009) | $cm$ (water equivalent thickness) |



**Table A2.** References for all observational datasets mentioned in Table 1

| name | available variables | reference |
|---|---|---|
| SYNOP stations | many, most prominently 2-meter temperature | e.g. Lawrimore et al. (2011) |
| FLUXNET stations | many, see Table 2 in reference | Pastorello et al. (2020) |
| E-OBS | 2-meter temperature, precipitation | Cornes et al. (2018) |
| GPM | precipitation | Huffmann et al. (2019) |
| CRU | 2-meter temperature, precipitation | Harris et al. (2020b) |
| ESA-CCI-SM | surface layer soil moisture | Gruber and Scanlon (2019), Dorigo et al. (2017), Gruber et al. (2017) |
| ISMN | soil moisture | Dorigo et al. (2011), Dorigo et al. (2013) |
| WECANN | latent heat, sensible heat | Alemohammad et al. (2017) |
| GLEAM | latent heat | Martens et al. (2017) |
| GSIM | runoff | Do et al. (2018) |
| G-RUN | runoff | Ghiggi et al. (2019) |
| GRACE | terrestrial water storage | Swenson (2012), Landerer and Swenson (2012), Swenson and Wahr (2006) |
| CERES | shortwave radiation, longwave radiation | Doelling (2017) |
| MODIS | ground temperature | Wan et al. (2015) |
| SEVIRI | ground temperature | Trigo et al. (2015) |





*Author contributions.* VB, LG, and SIS designed the study based on an initial idea from LG. SIS and LG secured the funding. VB and LG developed the framework and the evaluation. VB carried out the formal analysis and drafted the text. All authors contributed to reviewing
510  and editing the article.

*Competing interests.* The authors declare that they have no conflict of interest.

*Acknowledgements.* The authors would like to thank Nicolai Meinshausen for input on the initial idea, Mathias Hauser for help in publishing the accompanying python package and Martin Hirschi for post-processing the ERA5 data. This work was supported by ETH Research Grant ETH-08 19-1 (Data science for Integrating Complex Earth system observations, DICE) and ESA Climate Change Initiative for Soil Moisture
515  (Contract No. 4000126684/19/I-NB). Some of the calculations were performed using the Euler cluster at ETH Zurich. We would like to thank the ECMWF for creating and providing the ERA5 reanalysis product. Lastly, we would like to thank Roberto Villalobos, Jonas Jucker, Joel Zeder and Johannes Senn for feedback on the draft.



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
