# Peer review of "CLIMFILL v0.9: A Framework for Intelligently Gap filling Earth Observations"

_Geoscientific Model Development, 2021_

## Author Comment (AC1)

**In the following response, the original review is shown in grey and our response in green.**

**CEC1: 'Comment on gmd-2021-164', Astrid Kerkweg, 21 Jul 2021**

The main paper must give the model name and version number (or other unique identifier) in the title.

We thank the Executive Editor for this comment and adjust the title of the manuscript according to the GMD guidelines. We will add the version number to the title and change it accordingly.

---

## Author Comment (AC2)

**In the following response, the original review is shown in grey and our response in green.**

**RC1: ['Comment on gmd-2021-164'](), Anonymous Referee #1, 18 Aug 2021**

Review of « CLIMFILL: A Framework for Intelligently Gap-filling Earth Observations » by V. Bessenbacher et al.

This manuscript addresses an important problem: the gap-filling of global observations and the generation of continuous spatial and temporal data. It is well written and the methodology is mostly clear. This said, I have some reservation regarding the justification of the methodology and the validation approach. These are detailed below.

We thank Anonymous Referee #1 for this useful feedback and the helpful comments below.

Major comments:

- The method is well described, but involves a number of modeling choices (initial interpolation method, clustering approach, random forest estimation and averaging) that are not always justified, except by the experimental results showing that "it works". The problem is that I cannot make sure that it works with the current benchmarking. Indeed, the proposed method is compared only against the interpolation of step 1 of the proposed method itself. It is not in my opinion a sufficient benchmark. While it shows that steps 2-4 do have some added value compared to the extremely simple interpolation of step 1, added value against other interpolation methods is not demonstrated. By construction, it is expected that steps 1-4 perform than step 1 alone. I suggest to demonstrate the performance of the proposed approach is to compare it against something slightly more sophisticated, and already known to work in such contexts, for example (co-)kriging, possibly with a separate variogram model in each of the clusters defined in step 3.

We thank Anonymous Referee #1 for this valuable comment. Indeed the proposed interpolation method which is used in step 1 of CLIMFILL as well as as a benchmark for the CLIMFILL algorithm is simple. In step 1 a simple method was chosen deliberately since the primary focus is to initialize the subsequent iterative optimisation process. Moreover, the method was chosen to ensure that the problem stays computationally feasible. For the revised manuscript however, we will consider the application of other interpolation methods in step 1 of CLIMFILL as well as in benchmarking the CLIMFILL algorithm with other interpolation approaches such as kriging, Gaussian processes, or derivatives thereof.

Note also, that the high spatial and temporal resolution of the data and the relatively large number of observed values (as compared to an interpolation based on station data, like in E-OBS (Haylock et al, 2008)) make kriging computationally very expensive. Kriging has a cubic computational complexity, which means the computational expense increases with $O(n^{**}3)$, where n is the number of observed values. Standard kriging is

therefore not an option for our dataset which consists of several hundred thousand observed points per day.

To counteract this issue we are currently exploring recent literature on Gaussian Processes for large data, focussing e.g. on divide-and-conquer approaches in which the data are strategically split into more manageable chunks.

- The introduction stresses, with reason, that the reproduction of the dependencies between variables is critical, and that these dependencies are complex. However, the evaluation metric used relies on the assumption that these distributions are Gaussian, whereas it is clearly not the case, as seen in figure 6. Instead of eq. 2, I suggest using a metric that considers a numerical description of the joint distribution, such as for example the Jensen-Shannon divergence (or many other possible divergences available in the literature). Applied to the distributions in figure 6, the computational cost would be minimal.

For the revised manuscript, we will seek to replace the Bhattacharrya distance with a metric that does not assume Gaussian distributions. We will particularly look into options that make use of the distributions in Figure 6, once they are normalised (as suggested in the next comment). We thank the reviewer for suggesting the Jensen-Shannon divergence/distance, which we will consider.

- Some elements in figure 6 do not allow me to fully evaluate the results of the proposed method. I see significant differences between the distributions, e.g. in d) there is an important bias towards values of soil moisture around 0.3, which seems more important than in c). More generally, the distribution in d) looks globally like c) but smoothed (comparable to the smudging effect of adding a random noise). Similarly, in figure A1 there are important artifacts in the reproduction of the marginal distributions by CLIMFILL, which imply that the joint distribution is also inaccurate. Visually interpreting these effect is difficult because the joint distributions are presented as histograms, with counts of data instead of densities of probability. As a result, the integral of the joint distributions is not the same, especially for b). These histograms should be normalized by their integrals to reflect probabilities rather than counts.

We agree that normalising the histograms is a good idea and will do this for the revised manuscript. For the other issues mentioned in this part, please see the summarised response to a related comment below (comment starting with "While I see the logic in separating ….").

- In figure 11 as well as in the supplementary figures, I cannot see that CLIMFILL is systematically better than the simple interpolation. A quantitative assessment might help highlighting such differences. Why not using the same regions in figure 12 as in figure 11? Is the focus on randomly chosen regions or on the areas of larger discrepancies?

In Figure 11, the current SREX regions have been chosen to highlight exemplary how CLIMFILL performs in different climates across the globe. In Figure 12, the aim is to

focus on the regions where the largest difference is seen between original data, benchmark and CLIMFILL. We will revise these two figures to also quantitatively reflect the differences between benchmark and CLIMFILL.

- While I see the logic in separating the interpolation of a global trend (step 1) and detailed data-driven smaller scale features (steps 2-4), step 1 is a spatial KNN, which inherently assumes smoothness. This is a modeling decision having implications that are not evaluated. For example, the distributions in figure 6c and 6d present features that are absent from the dataset used to interpolate from (figure 6b). It means that some unobserved statistical properties have been created. It seems to me that the large peak in figure 6c and its smoother version in figure 6d are typical of nearest-neighbor algorithms that propagate a single nearest value far from observations.

We thank the reviewer for this helpful feedback. Indeed, the distribution in Figure 6c presents a feature that is absent in Figure 6b, namely a large number of previously missing surface layer soil moisture values that have been gap-filled with a value of roughly 0.3 m^3/m^3. The feature is a consequence of large gaps, where the initial gapfill by interpolation because no spatially or temporally close points are found to inform about the missing value. In these instances missing values are initialized with the (constant) global mean, which is causing this issue.

As mentioned above we will consider other interpolation techniques in the revised submission, which may alleviate this issue. In case this issue remains present (note e.g. that kriging would also converge to constant values in the presence of large gaps) we will ensure that it is properly discussed in the revised text.

- The interpolation approach is largely driven by the large number of features, which is fine and quite usual, but this means that it may not perform well in case of large gaps, or when some of these covariates are unknown. How does it perform when no covariates are present (or only e.g. topography and lat/lon)? Furthermore, it is mentioned in l. 155 and below that a shortcoming of existing gap-filling approaches is that they heavily rely on covariates and not on spatial relationships. As I understand it, CLIMFILL also relies largely on covariates (steps 2 and 3), and very little on spatial relationships (spatial dependence is only considered in step 1, and in a very loose way as through a nearest neighbor approach).

We thank the reviewer for pointing out that this part of the algorithm needs clarification in the text.

For our response below we assume that "the interpolation approach" refers to the whole CLIMFILL framework. For understanding the properties of CLIMFILL (and the imputation methods it is based on (Stekhoven and Bühlmann, 2012)), it is necessary to recall that the gap-filling happens in two distinct steps:

First, an ad-hoc estimate for missing values is derived for each variable separately. Here this is done using spatial interpolation and hence does not depend on the presence of gap-free covariates.

Second, the ad-hoc estimate of all considered variables is optimised using an iterative procedure by accounting for the covariance of all participating variables.

Consequently, the entire procedure is designed to operate on multivariate datasets in which gaps may be present in all variables. In particular, the second step can help to overcome limits of simple interpolation in the presence of large gaps, since information of other - possibly gap-free - variables is exploited. Nonetheless, we acknowledge that this may not always be the case and will revise the article to better communicate the merits and limitations of the presented approach.

- One problem I see with the proposed approach is that it does not consider or attempt to quantify uncertainty. The values in the middle of a large gap are given with the same confidence as for a single pixel gap. Similarly, the uncertainty should be larger when few covariates are present. Furthermore, on l.232 it is mentioned that the different clusterings obtained (which may in some sense convey a sense of variability) are averaged, thus collapsing any uncertainty into a mean value.

We fully agree with Anonymous Referee #1 that uncertainty quantification is an important aspect of statistical inference and that the confidence in the estimate is likely proportional to the density of available observations. However, with the structure of the considered data and the used methodology, this uncertainty cannot be treated using standard methods. For example,the algorithm consists of 4 steps, each carrying its specific methodological uncertainties and it is not straight-forward to combine these uncertainties. Furthermore, the complex dependence structure of the data, exhibiting temporal, spatial, and cross-variable dependence needs to be considered.

Consequently the development of well-behaved uncertainty estimates for CLIMFILL would require a significant and independent research effort which is beyond the scope of this study.

Minor/editorial comments:

- Some of the references are quite outdated, such as Rubin (1976) that is mentioned repeatedly, whereas the literature on spatial statistics and geostatistics, which is precisely concerned with interpolation in similar spatio-temporal applications, is quite incomplete. Some starting points could be:

Cressie, N. and K. Wikle (2011). Statistics for Spatio-Temporal Data. New Jersey, Wiley.

Chilès, J. P. and P. Delfiner (2012). Geostatistics: Modeling Spatial Uncertainty: Second Edition.

We thank the reviewer for these suggestions and will adapt the literature overview of spatial statistics and geostatistics to reflect the recent research, starting with the publications mentioned.

- Limitations of Gaussian processes are mentioned in l. 136, but with no details. There are several applications in the literature where Gaussian processes (or

other forms of random processes) have been successfully used with large datasets.

This is a valid point, we will revise this section to reflect this.

- 139: are well suited

The wording is changed accordingly.

- 153: provided by other variables

The wording is changed accordingly.

- Table 2, caption: it is not clear to me what is meant by "method class" in this table

We changed the Table caption to: "Main CLIMFILL settings per step. Each task can be performed using alternative methods mentioned in the last column"

- 170: outlook for possible future work

The wording is changed accordingly.

- Caption of figure 2: the framework is divides into four steps

The wording is changed accordingly.

- 203: constant maps describing properties

The wording is changed accordingly.

- 206: Please develop the motivation for the Takens Theorem. I do not see the link to the present approach, especially given the observational uncertainties considered here.

This section will be revised so that the link becomes more clear.

- 217: built from variables

The wording is changed accordingly.

- 216-218: This sentence seems to refer to the data format in the specific implementation described. Probably not needed in this methodological description.

We omit this sentence.

- 244: I understand that the subscript "updated" stands for estimated value. A more common notation would be to use a hat.

Thanks for this really useful suggestion. We will implement the notation with the hat.

- 244: the meaning of subscript m is unclear to me. Is it the same as in l. 217?

The meaning of the subscript "m" is the same in line 244 and line 217. It refers to the number of features, as described in the caption of Algorithm 1. We thank the reviewer for pointing out that this could be communicated a bit clearer. We will make sure to mention the definition of the subscripts m and n in the text before line 217.

- 266: it is mentioned that the proposed approach could be used to interpolate sparse in-situ measurements. This could be expanded upon or removed, as I do not see how it could be achieved easily in the present form because the model is heavily data-driven. Same comment for l.489-490.

This is a good idea. We will explore whether these two sections can be removed.

- Figure 5: why id the temporal window smaller in the future period than in the past?

We thank Anonymous Referee #1 for this helpful feedback. As a response to this comment and a comment from Referee #2 (Rene Orth), we will systematically explore the impact of different versions of feature selection, i.e. step 2 of the algorithm. We aim to quantitatively explore the effects of adding more features or removing features from the dataset.

- 305: the point is filled

The wording is changed accordingly.

- 323: are scalable

The wording is changed accordingly.

- 339: what is the criterion allowing to state that the shape of the distribution is well recovered?

When introducing quantitative measures of the reconstruction of the original distribution in the benchmark and CLIMFILL results, for example by using the Jensen Shannon divergence as proposed on the histograms in Figure 6. The text will be revised accordingly

- Legend of Figure 8: a) is described twice in the legend and c) is missing.

We replaced the respective sentence with "Comparison of artificial (a) random and (b) swaths-only missingness and (c) missingness in the real data"

- Legend of Figure 8: "In swaths-only…": incomplete sentence

We have added the missing part such that the sentence now reads "In swaths-only missingness we create long ellipses centered around the equator to simulate characteristic satellite swath missingness patterns."

- 361: Leads to…: incomplete sentence.

changed to "This leads…"

- 401: This last sentence is intriguing, especially during the presentation of results. It could be expanded upon in the discussion.

The mentioning of the bias reduction step will be revised.

- 468: recovering the physical

The wording is changed accordingly.

- 498-500: "closing the largest gaps first": it is not immediately clear to me that this would be the best strategy. One potential drawback of this approach might

be (but I am not sure) to artificially reduce the uncertainty related to the large gaps, precisely where uncertainty is important. One could also argue that a strategy could be to start with areas that are fairly certain (i.e. small gaps).

We acknowledge that the wording of this sentence is not optimal and will revise the text with focus on communicating uncertainties of the estimates.

- In figure A1, I recommend showing the precipitations on a log axis, and normalizing the joint distributions rather than displaying counts (same comment as for figure 6).

The figure is changed accordingly.

References:

Haylock, M.R., Hofstra, N., Klein Tank, A. M. G., Klok, E .J, Jones, P. D. and New, M. (2008): A European daily high-resolution gridded data set of surface temperature and precipitation for 1950-2006. Journal of Geophysical Research, doi:10.1029/2008JD010201

Stekhoven, D. J. and Bühlmann, P. (2012): MissForest–non-parametric missing value imputation for mixed-type data. Bioinformatics. 28, 112–118, doi:10.1093/bioinformatics/btr597

---

## Author Comment (AC3)

**In the following response, the original review is shown in grey and our response in green.**

**RC2: 'Review of "CLIMFILL: A Framework for Intelligently Gap-filling Earth Observations"', Rene Orth, 30 Aug 2021**

Review of Bessenbacher et al., gmd-2021-164

"CLIMFILL: A Framework for Intelligently Gap-filling Earth Observations"

This study introduces a sophisticated procedure to gap-fill Earth observation time series while benefitting from independently and concurrently observed related variables. The authors showcase the method with reanalysis data where some parts are intentionally masked, and the reconstructed estimates are finally compared with the original data. Thereby, they consider ground temperature, terrestrial water storage, surface layer soil moisture and precipitation and discuss the results both in terms of reconstucted individual time series, and for the interactions between reconstructed variables compared with respective estimates from the original data.
* * *
Recommendation:

I think the paper requires major revisions.

This is a useful and timely contribution for the Earth science community, and interesting for the readership of the Geoscientific Model Development. Benefitting from a growing suite of Earth observations,complex statistical tools and machine learning applications are increasingly employed in Earth science research.Mostly, these analysis tools require gap-free data which is often derived through gap-filling procedures.In this context, improving the quality of the gap-filling by exploiting the relationships between the independent Earth observations is a promising avenue.

However, I have some concerns regarding the description of the method and the benchmarking of the results, as detailed below.

We thank Rene Orth for this useful feedback and the helpful comments below.
* * *
General comments:

(1) Comparing the results from the plain interpolation with that at the end of all four steps of the gap-filling procedure is interesting to understand the method and the relevance of the various steps. However, it is not a suitable benchmarking exercise as it is to be expected that the results after four steps are closer to the original ERA5 data than the result after the first relatively crude interpolation step. Instead, an established univariate

gap-filling technique should be employed here as a benchmark to illustrate under which circumstances the presented methodology offers benefits over previous approaches. Also, this could reveal to which is extent the gap filling can be improved by (i) complete exploration of uni-variate time series beyond neighbors, versus (ii) a multivariate approach.

Thank you for pointing this out, a very similar issue was raised by Anonymous Referee #1 and we have copied the answer here for your convenience:

"We thank Anonymous Referee #1 for this valuable comment. Indeed the proposed interpolation method which is used in step 1 of CLIMFILL as well as as a benchmark for the CLIMFILL algorithm is simple. In step 1 a simple method was chosen deliberately since the primary focus is to initialize the subsequent iterative optimisation process. Moreover, the method was chosen to ensure that the problem stays computationally feasible. For the revised manuscript however, we will consider the application of other interpolation methods in step 1 of CLIMFILL as well as in benchmarking the CLIMFILL algorithm with other interpolation approaches such as kriging, Gaussian processes, or derivatives thereof.

Note also, that the high spatial and temporal resolution of the data and the relatively large number of observed values (as compared to an interpolation based on station data, like in E-OBS (Haylock et al, 2008)) make kriging computationally very expensive. Kriging has a cubic computational complexity, which means the computational expense increases with O(n**3), where n is the number of observed values. Standard kriging is therefore not an option for our dataset which consists of several hundred thousand observed points per day.

To counteract this issue we are currently exploring recent literature on Gaussian Processes for large data, focussing e.g. on divide-and-conquer approaches in which the data are strategically split into more manageable chunks."

(2) I think it would be useful for future CLIMFILL users to give more guidance on the methods to use in each step of the algorithm. Table 2 offers many possible choices, but in addition some recommendations would be needed on when to use which method and why. Also, the selection of employed variables is important as their inter-relations are a key source for the gap reconstructions, so also some additional advice on this would be helpful.

Thank you for this suggestion. We will include additional text discussing the merits of methodological choices, while also acknowledging that the perfect setup will likely depend on the application. For the selection of employed variables, please see our reply to your next comment (3).

(3) I think that the feature selection is a bit arbitrary and dependent on expert knowledge. To somewhat address this issue, maybe several features could be used by default, such as the 34 features used in the presented example and maybe even additional time lags and windows. Then, the random forest model  can be employed to rank the features by their importance (e.g. using SHAP value importance) to make  a more informed decision

on the useful features. Finally, the gap-filling could be re-run with only retaining relevant features.

This is a very helpful comment. Anonymous Referee #1 also suggested adding more time lags and windows, especially in the future. The initial choice indeed was indeed guided by expert knowledge. For the revised version we aim to quantitatively quantify feature importance and explore the effects of adding more features or removing features from the dataset.

(4) There is advanced statistical and data science language used across the manuscript and I recommend to clarify this with additional information to allow a broader geoscientific audience to follow this manuscript. Please see my respective suggestions in the specific comments below.

We will revise the mentioned sections of the text to make it more accessible to a broader geoscientific audience.

I do not wish to remain anonymous - Rene Orth.
* * *
Specific comments:

line 2: estimates for what?

changed or "estimates of statistical moments"

line 5: remove "up"

The wording is changed accordingly.

line 7: I agree that technically the algorithm does not require a gap-free donor variable; however if all variables have gaps at the same time and if this period is longer, then the final gap-fill estimate will naturally have a low quality

Yes, it is true that if all variables have gaps at the same time this will likely affect the quality of the estimate. Note, however, that this does not have to result in poor overall performance in case of a good first guess in step 1.

line 15: "profit", maybe rephrase as "are improved by"

The wording is changed accordingly.

lines 45, 144 & Table 1: Jung et al. 2019 and O & Orth 2021 are relevant studies in this context and could be mentioned here

We thank the reviewer for pointing us towards these publications.

line 46: please clarify "scale somewhere between"

We suggest replacing the mentioned text with "are positioned on a gradient between"

line 84: please clarify "difficult observational record"

We change the wording from "difficult" to "fragmented" in coherence with line 28, 43 and 51

lines 108/109 and 111 are in contrast to each other

A good point. We will either describe "faulty sensor pixel" scenario as MCAR scenario or remove sentence "All these three missingness patterns can be observed in earth observation data:"

line 151: this is unclear, please rephrase

Sentence changed to "There is a growing body of literature of different methods that are originally equipped with dealing with only spatial or temporal relations *that* are expanded and altered to take into account the information from the other dimension as well (von Buttlar et al., 2014; Gerber et al., 2018)."

line 154: "another" should be "other" I guess

The wording is changed accordingly.

Table 2, caption: "other" should be "another" I guess

The wording is changed accordingly.

Table 2, right column: "or more complex interpolation methods", "Guided by ...", these are not exactly examples as the column title suggests

The wording is changed to "Possible alternative methods"

line 170: remove "on"

The wording is changed accordingly.

line 171: feels a bit random which letters are capitalized here and which are not

The section title will be changed to "CLIMFILL: A Generalised Framework for Infilling Missing Values in Multivariate Spatio-Temporal Geoscientific Data"

line 173: "the highly structured nature", please explain

We suggest replacing "the highly structured nature" with "the spatial, temporal and cross-variable dependence structure" to make the meaning more clear.

Figure 2, caption: The framework is divided into four steps, not three.

This will be corrected.

line 178: Abbreviation CLIMFILL is mentioned earlier and should be explained at the first occasion

The abbreviation CLIMFILL will only be explained in the abstract at the first mention.

line 181: please clarify "correlation structure"

We will replace "correlation structure" with "dependence structure" to ensure consistency across the text. (for example, lines 189, 366, 450,...)

lines 203, 311: please clarify "constant"

The wording will be changed from "constant" to "time-independent"

line 216: quotation marks not needed

The sentence is changed accordingly.

lines 229: please clarify "stabilising the results"

The sentence now reads: "For reducing the risk of discontinuities at the cluster edges, the clustering procedure is repeated E times with different numbers of terminal clusters on copies of the data X(1),...,X(E)."

line 231: please clarify "terminal clusters"

We will remove the word "terminal" for clarification.

line 243: I think this should be "to overwrite the former estimates"

The wording is changed accordingly.

lines 250/251: "learns different weights", please clarify

We replace "weights" with "model parameters" for clarification.

Figure 3, caption: replace "substracting" with "subtracting"

The wording is changed accordingly.

line 272: How are deserts defined and detected?

We change the sentence from "Permanently glaciated areas and deserts are masked" to "Permanently glaciated areas and deserts (defined as areas with less 50 mm average yearly precipitation in the years 2003-2012) are masked".

line 311: It should be 4 and not 3 additional features I guess?

We replace "3" with "four".

line 314: please clarify "non-normality"

We change "treat its inherent non-normality" to "transform the values into a gaussian distribution".

line 316: How does this add up to 34?

four variables + precipitation additional variable = 5

four embedded features per variable = 4*5 = 20

6 constant maps

lat, lon time = 3

5 + 20 + 6 + 3 = 34 features

We will add in the text brackets to ensure the number of features is clarified. Please also note that these might change due to comments of both reviewers about feature selection and feature importance (see above).

line 319: "respectively" should be added after "clusters" I guess

The wording is changed accordingly.

line 326: I wonder if and how different spatial resolutions can affect the accuracy of the gap filling, it would be great if the authors could shortly discuss this.

This is a good question. The different spatial resolutions indeed have similar accuracy in gapfilling, and we will add a discussion and a figure in the appendix to show this.

line 326: "where one fold is one year", please clarify

Thanks for spotting this example of convoluted machine-learning lingo. We change the sentence from "The hyper-parameters 325 of the supervised learning functions are determined via leave-one-out cross-validation on clustered ERA5 data between 2015 and 2020 downscaled to 2.5 degrees resolution, where one fold is one year." to "The hyper-parameters 325 of the supervised learning functions are determined via leave-one-year-out cross-validation on clustered ERA5 data between 2015 and 2020 downscaled to 2.5 degrees resolution."

Figure 7, caption: what is "CLIMPUTE-RF"?

Thanks for spotting this typo. We change it to "CLIMFILL-RF"

line 351: please clarify "det"

$\det$ is the determinant of the respective covariance matrix. Note that sentence however might disappear when the Bhattacharyya distance is replaced in response to a suggestion of Anonymous Referee #1 .

Figure 8, caption: sentences should not end with "with" and "create".

The wording is changed accordingly.

line 361: "This" should be added before "leads".

The wording is changed accordingly.

line 367, section 3.4: I very much like the idea of studying the performance of the gap-filling across missingness patterns and different severity of the gaps.

We thank the reviewer very much for this feedback.

Figure 10, caption: the B-distance is not actually displayed in this figure

We change from "pearson correlation, root mean square error (RMSE) and B-distance (for more details see text)" to "pearson correlation and root mean square error (RMSE)"

line 373: How exactly are the satellite swaths imitated?

We have added an explanation in caption of figure 8 as a response to Referees #1 and #2: "In swaths-only missingness we create long ellipses centered around the equator to simulate characteristic satellite swath missingness patterns".

line 401: I do not quite understand the point on the bias correction.

The mentioning of the bias reduction step will be revised.

line 427: similar in "remotely sensed" data but underestimated in "satellite observations", this should be the same thing?

We agree this sentence reads a bit confusing and will revise it accordingly.

Figure 2: The figure is rather small now and should be enlarged to make it easier to see all details.

The figure is changed accordingly.

Figure 4: The months axis should not go to 12.5

The figure is changed accordingly.

References:

Jung, M., et al., The FLUXCOM ensemble of global land-atmosphere energy fluxes, Sci. Data 6, 74 (2019).

O, S. and R. Orth, Global soil moisture data derived through machine learning trained with in-situ measurements, Sci. Data 8, 170 (2021).

References:

Haylock, M.R., Hofstra, N., Klein Tank, A. M. G., Klok, E .J, Jones, P. D. and New, M. (2008): A European daily high-resolution gridded data set of surface temperature and precipitation for 1950-2006. Journal of Geophysical Research, doi:10.1029/2008JD010201

---

## Author Response (AR1)

**Reply to Reviewer Comments to the Manuscript**

**CLIMFILL v0.9: A Framework for Intelligently Gapfilling Earth Observations**

Verena Bessenbacher, Sonia I. Seneviratne, Lukas Gudmundsson Geophysical Model Development, doi:10.5194/gmd-2021-164

**RC:** *Reviewer Comment*, AR: *Authors response*,  $\Box$  Manuscript text

**1. Letter to the editor**

Dear Prof. Tomomichi Kato,

Please find enclosed the revised version of the article entitled "CLIMFILL v0.9: A Framework for Intelligently Gap filling Earth Observations" (Paper GMD-2021-164).

Overall, the reviewers did support the methodological basis of the article. However, both reviewers suggested some modifications to parts of the framework. We have addressed them below in our point-by-point response letter. Following these comments, changes have been made to the individual steps of the gap-filling process as well as to its evaluation. In particular, the following main revisions were made to the manuscript:

- In the first step of the algorithm, the simple spatiotemporal interpolation using nearest neighbours was replaced by a more sophisticated, state-of-the art interpolation that is based on a well established approach from Haylock et al., (2008) with modifications to reduce computational expense. It is a combination of thin-plate-spline interpolation and kriging. Consequently, also the benchmark used for comparing CLIMFILL has changed to this interpolation method.
- We replaced the Bhattacharyya-distance with a non-parametric measure for computing the distance of two multivariate distributions, the Jenson-Shannon distance, as suggested by Anonymous Referee #1, to account for the non-normality of the data.
- In the second step of the algorithm, several "flavours" of feature engineering have been explored to analyse the impact of considering different sets of variables to the gap-filling. We evaluate which set of features gives the best results at the beginning of the results section.

With these modifications triggered by the suggestions of the reviewers, the computational expense grew outside of what was capable for our computational resources. Therefore, we had to decrease the amount of data used in this study from ten years to one year. Most prominently, trying different sets of features requiring running CLIMFILL several times as compared to before, only once. One CLIMFILL run on our machines with 128 cores takes approx 24h with the newly reduced dataset. Furthermore, computing the Jenson-Shannon distance is approx. 100 times more expensive than the Bhattacharyya-distance. The caveat of this decision is that the analysis of interannual variability is inhibited, which triggered changes in Figure 11 and 12 consequently.

Note, however, that neither the modifications to the methodology suggested by the referees nor decrease in the amount of data have changed the overall conclusions of the manuscript.

The valuable comments from the reviewers changed the structure of article substantially throughout the text. These changes can be viewed in the accompanyed difference-document. We are confident that the above-mentioned revisions of the paper together with the updated supporting information have increased the value of the submitted manuscript.

Yours Sincerely,

Verena Bessenbacher (on behalf of all co-authors)

**2. Astrid Kerkweg**

- 2.1. Comment
- RC: The main paper must give the model name and version number (or other unique identifier) in the title.
- AR: We thank the Executive Editor for this comment and adjusted the title of the manuscript according to the GMD guidelines.

CLIMFILL v0.9: A Framework for Intelligently Gap-filling Earth Observations

**3. Anonymous Referee #1**

- RC: Review of « CLIMFILL: A Framework for Intelligently Gap-filling Earth Observations » by V. Bessenbacher et al. This manuscript addresses an important problem: the gap-filling of global observations and the generation of continuous spatial and temporal data. It is well written and the methodology is mostly clear. This said, I have some reservation regarding the justification of the methodology and the validation approach. These are detailed below.
- AR: We thank Anonymous Referee #1 for this useful feedback and the helpful comments below.
- RC: The method is well described, but involves a number of modeling choices (initial interpolation method, clustering approach, random forest estimation and averaging) that are not always justified, except by the experimental results showing that "it works". The problem is that I cannot make sure that it works with the current benchmarking. Indeed, the proposed method is compared only against the interpolation of step 1 of the proposed method itself. It is not in my opinion a sufficient benchmark. While it shows that steps 2-4 do have some added value compared to the extremely simple interpolation of step 1, added value against other interpolation methods is not demonstrated. By construction, it is expected that steps 1-4 perform than step 1 alone. I suggest to demonstrate the performance of the proposed approach is to compare it against something slightly more sophisticated, and already known to work in such contexts, for example (co-)kriging, possibly with a separate variogram model in each of the clusters defined in step 3.
- AR: We thank Anonymous Referee #1 for this valuable comment. Indeed the proposed interpolation method which is used in step 1 of CLIMFILL as well as as a benchmark for the CLIMFILL algorithm is simple. In step

1 a simple method was chosen deliberately since the primary focus is to initialize the subsequent iterative optimisation process. Moreover, the method was chosen to ensure that the problem stays computationally feasible. Due to the high spatial and temporal resolution of the data and the relatively large number of observed values, kriging is computationally very expensive. Kriging has a cubic computational complexity, which means the computational expense increases with  $O(n^{**3})$ , where n is the number of observed values. Standard kriging is therefore not an option for our dataset which consists of several hundred thousand observed points per day. For the revised manuscript however, we have now replaced both the interpolation in step 1 and the benchmark with a state-of-the-art spatial interpolation method borrowed from Haylock et al., (2008) adapted with a sparse, computationally efficient kriging method Das et al., (2018) to make the problem computationally feasible. We described the procedure in the methods section:

The interpolation step creates initial estimates based on the spatial or spatiotemporal context of the gap using interpolation. Following the approach of Haylock et al. (2008), the data is first divided into monthly climatology maps and anomalies. The climatology maps are gap-filled using thin-plate-spline interpolation to represent the spatial trends in the data. Subsequently, the daily anomalies from the monthly climatology are gap-filled using kriging. In contrast to the E-OBS dataset created in (Haylock et al., 2008) from in-situ observations, satellite data has a much larger number of observed values, making a direct implementation of this approach computationally infeasible. For the interpolation of the monthly climatology maps we there- fore restrict the thin-plate-spline interpolation to the 50 closest neighbors of each point. The interpolation of the daily anomalies follows Das et al. (2018), who suggest reducing complexity of kriging/Gaussian Process regression by repeated interpolations on random sub-samples of all available data points and averaging the resulting estimates. In particular, the missing values in the anomalies are estimated by randomly selecting 1000 observed points per month over which the interpolation is calculated. This is repeated five times and the mean of all interpolations for each missing point is taken as the gap-fill estimate. Finally, monthly maps and anomalies are summed up to form the initial gap-fill estimate from step 1.

Please also note changes to the framework regarding to the clustering step described in the answers below.

- RC: The introduction stresses, with reason, that the reproduction of the dependencies between variables is critical, and that these dependencies are complex. However, the evaluation metric used relies on the assumption that these distributions are Gaussian, whereas it is clearly not the case, as seen in figure 6. Instead of eq. 2, I suggest using a metric that considers a numerical description of the joint distribution, such as for example the Jensen-Shannon divergence (or many other possible divergences available in the literature). Applied to the distributions in figure 6, the computational cost would be minimal.
- AR: We thank Anonymous Referee #1 for this valuable comment. For the revised manuscript, we have replaced the B-distance with the Jenson-Shannon distance based on the distributions in Figure 6. We describe this in the text:

We additionally examine which subset of features is most descriptive for the problem at hand and settle on one of the propositions. To allow for a quantitative assessment of the similarity of the multi-variate distributions of observed and simulated variables. To overcome this issue, we apply a scalar measure of multivariate similarity. In this study, we use the Jenson-Shannon distance (JS-distance). This measure compares the multivariate distance between two datasets or multivariate distributions, where a value of one means that the two samples are from the same distribution, and a positive value indicates one indicates that the distributions are not overlapping. We apply the JS-distance on the four-dimensional histograms computed of the relative distance between two distributions .four variables using 50 bins for each variable.

- RC: Some elements in figure 6 do not allow me to fully evaluate the results of the proposed method. I see significant differences between the distributions, e.g. in d) there is an important bias towards values of soil moisture around 0.3, which seems more important than in c). More generally, the distribution in d) looks globally like c) but smoothed (comparable to the smudging effect of adding a random noise). Similarly, in figure A1 there are important artifacts in the reproduction of the marginal distributions by CLIMFILL, which imply that the joint distribution is also inaccurate. Visually interpreting these effect is difficult because the joint distributions are presented as histograms, with counts of data instead of densities of probability. As a result, the integral of the joint distributions is not the same, especially for b). These histograms should be normalized by their integrals to reflect probabilities rather than counts.
- AR: We have normalised the histograms by dividing the counts by the number of datapoints in the original ERA5 data to ensure comparability among the plots (a) through (d). For the other issues mentioned in this part, please see the summarised response to a related comment below (comment starting with "While I see the logic in separating ....").
- **RC:** In figure 11 as well as in the supplementary figures, I cannot see that CLIMFILL is systematically better than the simple interpolation. A quantitative assessment might help highlighting such differences. Why not using the same regions in figure 12 as in figure 11? Is the focus on randomly chosen regions or on the areas of larger discrepancies?
- AR: We have changed Figure 11 as follows: we have added the RMSE of original and CLIMFILL values to allow for an overview across regions and highlight several regions to show exemplarly how CLIMFILL works there. Figure 12 has changed completely and therefore this comment is not applicable anymore.
- RC: While I see the logic in separating the interpolation of a global trend (step 1) and detailed data-driven smaller scale features (steps 2-4), step 1 is a spatial KNN, which inherently assumes smoothness. This is a modeling decision having implications that are not evaluated. For example, the distributions in figure 6c and 6d present features that are absent from the dataset used to interpolate from (figure 6b). It means that some unobserved statistical properties have been created. It seems to me that the large peak in figure 6c and its smoother version in figure 6d are typical of nearest-neighbor algorithms that propagate a single nearest value far from observations.
- AR: We thank the reviewer for this helpful feedback. Indeed, the distribution in Figure 6c presented a feature that was absent in Figure 6b, namely a large number of previously missing surface layer soil moisture values that have been gap-filled with a value of roughly  $0.3 \frac{m^3}{m^3}$ . The feature was a consequence of large gaps, where the initial gapfill by interpolation because no spatially or temporally close points are found to inform about the missing value. In these instances missing values are initialized with the (constant) global mean, which is

causing this issue. Replacing the interpolation step with the new method described above has now removed these statistical artefacts.

- RC: The interpolation approach is largely driven by the large number of features, which is fine and quite usual, but this means that it may not perform well in case of large gaps, or when some of these covariates are unknown. How does it perform when no covariates are present (or only e.g. topography and lat/lon)? Furthermore, it is mentioned in l. 155 and below that a shortcoming of existing gap-filling approaches is that they heavily rely on covariates and not on spatial relationships. As I understand it, CLIMFILL also relies largely on covariates (steps 2 and 3), and very little on spatial relationships (spatial dependence is only considered in step 1, and in a very loose way as through a nearest neighbor approach).
- AR: When no covariates are present, the initial interpolation gap fill is not changed by the iterative, multivariate procedure in step 4 and therefore the initial gap fill is used. We have rewritten the literature review to stress that the shortcomings of existing gap filling approaches is that they cannot ingest spatial, temporal and multivariate dependencies once missing values are present in more than one of the included observations and hope this clarifies this issue. Please see the new introduction text for the respective changes, as there are too many to quote them here directly.
- RC: One problem I see with the proposed approach is that it does not consider or attempt to quantify uncertainty. The values in the middle of a large gap are given with the same confidence as for a single pixel gap. Similarly, the uncertainty should be larger when few covariates are present. Furthermore, on 1.232 it is mentioned that the different clusterings obtained (which may in some sense convey a sense of variability) are averaged, thus collapsing any uncertainty into a mean value.
- AR: We fully agree with Anonymous Referee #1 that uncertainty quantification is an important aspect of statistical inference and that the confidence in the estimate is likely proportional to the density of available observations. However, with the structure of the considered data and the used methodology, this uncertainty cannot be treated using standard methods. For example, the algorithm consists of 4 steps, each carrying its specific methodological uncertainties and it is not straight-forward to combine these uncertainties. Furthermore, the complex dependence structure of the data, exhibiting temporal, spatial, and cross-variable dependence needs to be considered. Consequently the development of well-behaved uncertainty estimates for CLIMFILL would require a significant and independent research effort which is beyond the scope of this study. Consequently, we have removed the part where different clusterings are obtained: instead of 3 different clusterings where each of them is gapfilled with 100 trees in step 4, we now have only one clustering, but each of them with 300 trees. We have verified that the results are similar with this changed method.
- **RC:** Some of the references are quite outdated, such as Rubin (1976) that is mentioned repeatedly, whereas the literature on spatial statistics and geostatistics, which is precisely concerned with interpolation in similar spatio-temporal applications, is quite incomplete. Some starting points could be:
  - Cressie, N. and K. Wikle (2011). Statistics for Spatio-Temporal Data. New Jersey, Wiley.
  - Chilès, J. P. and P. Delfiner (2012). Geostatistics: Modeling Spatial Uncertainty: Second Edition.
- AR: We thank the reviewer for the provided sources and have added literature on spatial statistics in the introduction:

In the geoscientific literature, among the most commonly used approaches for estimating unobserved points are spatial and temporal interpolation methods, including nearest neighbour regression as well as kriging and derivatives thereof (Liu et al., 45 2018; Cowtan and Way, 2014; Haylock et al., 2008; Cressie et al., 2006) (for an overview see Cressie and Wikle 2015; Allard et al. 2013). Spectral methods are used as well (Zhang et al., 2018; von Buttlar et al., 2014; Brooks et al., 2012).

- RC: Limitations of Gaussian processes are mentioned in l. 136, but with no details. There are several applications in the literature where Gaussian processes (or other forms of random processes) have been successfully used with large datasets.
- AR: Gaussian processes and their applications with large datasets are now reviewed in the introduction:

A wide range of algorithms that make use of cross-variable dependence to estimate missing values exist in statistical liter- ature. In the following, we are highlighting two common approaches: On one hand, Gaussian processes are a natural choice for gap filling problems (Gelfand and Schliep, 2016) and are mathematically identical to kriging, if the predictors are latitude and longitude. Gaussian processes however have limitations when moving to large data (Heaton et al., 2019) as is the case in Earth observation data. In recent years, some applications of Gaussian processes have been shown to work in settings with too much data to estimate the co-variance matrix between all datapoints precisely. They estimate the co-variance matrix via sophisticated sampling techniques (Wang and Chaib-draa, 2017; Das et al., 2018), pre-process the data via dimension reduction methods (Banerjee et al., 2008) or apply the Gaussian Process to local subsets of the data (Gramacy and Apley, 2015; Datta et al., 2016).

**RC: 139: are well suited**

AR: The wording is changed accordingly.

On the other hand, iterative procedures like the MICE-Algorithm ("Multiple imputation by chained equation", van Buuren (2018)) are suited well well suited for multivariate imputation and scale to large data, but cannot account for neighborhood relations.

**RC:** 153: provided by other variables**

- AR: The text has changed, therefore this comment is not applicable anymore.
- **RC:** Table 2, caption: it is not clear to me what is meant by "method class" in this table
- AR: The table has been removed, therefore this comment is not applicable anymore.
- **RC:** 170: outlook for possible future work
- AR: The wording is changed accordingly.

Finally, Sect. 4 discusses the results and provides a conclusion and an outlook on for possible future work.

**RC: Caption of figure 2: the framework is divides into four steps**

AR: The wording is changed accordingly.

The framework is divided into three four steps.

**RC:** 203: constant maps describing properties**

AR: The wording is changed accordingly.

For example, gap-free constant maps of time-independent maps describing properties of the land surface such as topography or land cover can be included.

- RC: 206: Please develop the motivation for the Takens Theorem. I do not see the link to the present approach, especially given the observational uncertainties considered here.
- AR: The reference to the Takens Theorem has been removed.
- **RC:** 217: built from variables
- AR: This sentence has been omitted as a response to the next comment.

For the application of data science methods, the data need to be rearranged in a table X build from variables  $v_1, ..., v_n$  and derived features  $v_1^*, ..., v_m^*$  as columns and space-time points as rows.

- **RC:** 216-218: This sentence seems to refer to the data format in the specific implementation described. Probably not needed in this methodological description.
- AR: We omit this sentence

For the application of data science methods, the data need to be rearranged in a table X build from variables  $v_1, ..., v_n$  and derived features  $v_1^*, ..., v_m^*$  as columns and space-time points as rows.

**RC: 244: I understand that the subscript "updated" stands for estimated value. A more common notation would be to use a hat.**

AR: Thanks for this really useful suggestion. We have implemented the notation with the hat in the text:

Subsequently both  $\mathbf{y}_{v}^{(e,k)}$  and  $\mathbf{X}_{-v}^{(e,k)}$  are divided into two sets of data points: (1) all data points where  $\mathbf{y}_{v}^{(e,k)}$  was originally observed are used to fit the supervised learning method  $\mathbf{y}_{v,o}^{(e,k)} = f(\mathbf{X}_{-v,o}^{(e,k)})$  and (2) all data points where  $\mathbf{y}_{v}^{(e,k)}$  was missing  $\mathbf{y}_{v,m}^{(e,k)}$  are predicted from the fitted function and to overwrite the former estimates:  $\mathbf{y}_{v,m}^{(e,k),updated} = f(\mathbf{X}_{-v,m}^{(e,k)})\mathbf{\hat{y}}_{v,m}^{(e,k)} = f(\mathbf{X}_{-v,m}^{(e,k)})$ .

and Algorithm 1:

| Create                                                                                                                                              | an | updated | estimate | with | the | fitted | regression | model |
|-----------------------------------------------------------------------------------------------------------------------------------------------------|----|---------|----------|------|-----|--------|------------|-------|
| $\mathbf{y}_{v,m}^{(e,k),updated} = f(\mathbf{X}_{-v,m}^{(e,k)}) \hat{\mathbf{y}}_{u,m}^{(e,k)} = f(\mathbf{X}_{-v,m}^{(e,k)}).$                    |    |         |          |      |     |        |            |       |
| Replace $\mathbf{y}_{v,m}^{(e,k)}$ with the new updated $\mathbf{y}_{v,m}^{(e,k),updated} \hat{\mathbf{y}}_{u,m}^{(e,k)}$ in $\mathbf{X}^{(e,k)}$ . |    |         |          |      |     |        |            |       |

**RC: 244: the meaning of subscript m is unclear to me. Is it the same as in l. 217?**

AR: The meaning of the subscript m is not the same in line 244 and line 217. We thank the reviewer for pointing out this inconsistency in our notation. In line 217, we are referring to the number of embedded features, in line 244 we refer to the part of the datapoints where the predictor variable is missing. We have changed the notation such that  $n_v$  is the number of variables (previously n) and  $n_f$  is the number of features (previously m) and m solely refers to the part of the datapoints where the predictor variable is missing. This changes the caption of Algorithm 1:

Pseudo-code algorithm of the CLIMFILL clustering and learning step (step 3 and 4), where E is the number of epochs, K is the number of clusters,  $n n_{y}$  is the number of variables and  $m n_{f}$  the number of features.  $\mathbf{X}_{-v}$  refers to the data table with all variables except v.

and line 1 in Algorithm 1:

```
X is a matrix containing all variables and features as n + m n_x + n_f columns and all data points as rows.
```

- RC: 26: it is mentioned that the proposed approach could be used to interpolate sparse in-situ measurements. This could be expanded upon or removed, as I do not see how it could be achieved easily in the present form because the model is heavily data-driven. Same comment for 1.489-490.
- AR: This is a very valid point. Indeed using the presented framework for interpolating sparse in-situ measurements is not immediately straight forward and some design choices would need to be made which are outside of the scope of this paper. The sections are therefore removed from the text.

We note however that the framework is naturally extendable to include more satellite observations, and in situ observations that can be treated as a very sparse gridded product.

in situ data could possibly be included as well if treated as a very sparsely gridded data where the area of representation for the point measurement is accessed (see e.g. Nicolai-Shaw et al., 2017).

**RC:** Figure 5: why id the temporal window smaller in the future period than in the past?**

AR: We thank Anonymous Referee #1 for this helpful feedback. As a response to this comment and a related comment from Referee #2 (Rene Orth), we have created symmetric embedded features for both the future and the past and systematically explore the impact of different versions of feature selection. For details, please refer to the detailed reply to the comment from Referee #2 below.

**RC: 305: the point is filled**

AR: This paragraph was rewritten due to the new interpolation method. This comment is therefore not applicable anymore.

**RC:** *323: are scalable**

AR: The wording is changed accordingly.

Random Forests have have favorable properties for gap filling applications: they can handle mixed types of data, are scale-able\_scalable to large amounts of data and non-parametric, i.e. adaptive to linear and non-linear relationships (Tang and Ishwaran, 2017).

**RC:** 339: what is the criterion allowing to state that the shape of the distribution is well recovered?**

AR: Quantitative measures of the recovery of the original distribution are included by using the Jensen Shannon divergence as proposed by the reviewer above. Please see Figure 6 and the accompanying text.

**RC:** Legend of Figure 8: a) is described twice in the legend and c) is missing.**

AR: The caption of Figure 8 is changed as follows:

Comparison of (a) artificial (a) random and (b) swaths-only missingness and (c) missingness in the real data in example snapshot of ERA5 ground temperature on 1st of August 2003 with. 2003.

**RC: Legend of Figure 8: "In swaths-only...": incomplete sentence**

AR: The caption of Figure 8 is changed as follows:

In swaths-only missingness we create long ellipses centered around the equator to simulate characteristic satellite swath missingness patterns. Note that the two missingness patterns are not exactly the same for each day and variable to allow for mutual learning.

**RC:** 361: Leads to...: incomplete sentence.**

AR: This paragraph was rewritten due to the new distance measure. This comment is therefore not applicable anymore.

**RC:** 401: This last sentence is intriguing, especially during the presentation of results. It could be expanded upon in the discussion.**

AR: Thank you for this comment. As a reaction to this comment and the one of Referee #2 regarding bias correction, we have decided to remove the mentioning of bias correction in the manuscript. It is not straight-forward to apply and this would be out of the scope of this manuscript.

Introducing an additional bias correction step could help alleviate these problems.

**RC:** 468: recovering the physical**

AR: The wording is changed accordingly.

Since the framework is targeted at recovering the the physical dependence structure across variables, the improvement in univariate measures like correlation and bias tend to be improved at a smaller scale than the multivariate dependence structure.

- RC: 498-500: "closing the largest gaps first": it is not immediately clear to me that this would be the best strategy. One potential drawback of this approach might be (but I am not sure) to artificially reduce the uncertainty related to the large gaps, precisely where uncertainty is important. One could also argue that a strategy could be to start with areas that are fairly certain (i.e. small gaps).
- AR: This part of the text has been removed, therefore this comment is not applicable anymore.
- **RC:** In figure A1, I recommend showing the precipitations on a log axis, and normalizing the joint distributions rather than displaying counts (same comment as for figure 6).
- AR: The joint distributions are normalised. However, we decided to not plot the precipitation in log axis because then the comparison between the distributions for no rain would not be possible.

**4. Rene Orth, Referee #2**

**RC: Review of Bessenbacher et al., gmd-2021-164**

"CLIMFILL: A Framework for Intelligently Gap-filling Earth Observations"

This study introduces a sophisticated procedure to gap-fill Earth observation time series while benefitting from independently and concurrently observed related variables. The authors showcase the method with reanalysis data where some parts are intentionally masked, and the reconstructed estimates are finally compared with the original data. Thereby, they consider ground temperature, terrestrial water storage, surface layer soil moisture and precipitation and discuss the results both in terms of reconstructed individual time series, and for the interactions between reconstructed variables compared with respective estimates from the original data.

**Recommendation:**

I think the paper requires major revisions. This is a useful and timely contribution for the Earth science community, and interesting for the readership of the Geoscientific Model Development. Benefitting from a growing suite of Earth observations, complex statistical tools and machine learning applications are

increasingly employed in Earth science research.Mostly, these analysis tools require gap-free data which is often derived through gap-filling procedures.In this context, improving the quality of the gap-filling by exploiting the relationships between the independent Earth observations is a promising avenue. However, I have some concerns regarding the description of the method and the benchmarking of the results, as detailed below.

- AR: We thank Rene Orth for this useful feedback and the helpful comments below.
- RC: Comparing the results from the plain interpolation with that at the end of all four steps of the gap-filling procedure is interesting to understand the method and the relevance of the various steps. However, it is not a suitable benchmarking exercise as it is to be expected that the results after four steps are closer to the original ERA5 data than the result after the first relatively crude interpolation step. Instead, an established univariate gap-filling technique should be employed here as a benchmark to illustrate under which circumstances the presented methodology offers benefits over previous approaches. Also, this could reveal to which is extent the gap filling can be improved by (i) complete exploration of uni-variate time series beyond neighbors, versus (ii) a multivariate approach.
- AR: Thank you for pointing this out, a very similar issue was raised by Anonymous Referee #1 and we have copied the answer here for your convenience:

The interpolation step creates initial estimates based on the spatial or spatiotemporal context of the gap using interpolation. Following the approach of Haylock et al. (2008), the data is first divided into monthly climatology maps and anomalies. The climatology maps are gap-filled using thin-plate-spline interpolation to represent the spatial trends in the data. Subsequently, the daily anomalies from the monthly climatology are gap-filled using kriging. In contrast to the E-OBS dataset created in (Haylock et al., 2008) from in-situ observations, satellite data has a much larger number of observed values, making a direct implementation of this approach computationally infeasible. For the interpolation of the monthly climatology maps we there- fore restrict the thin-plate-spline interpolation to the 50 closest neighbors of each point. The interpolation of the daily anomalies follows Das et al. (2018), who suggest reducing complexity of kriging/Gaussian Process regression by repeated interpolations on random sub-samples of all available data points and averaging the resulting estimates. In particular, the missing values in the anomalies are estimated by randomly selecting 1000 observed points per month over which the interpolation is calculated. This is repeated five times and the mean of all interpolations for each missing point is taken as the gap-fill estimate. Finally, monthly maps and anomalies are summed up to form the initial gap-fill estimate from step 1.

- RC: I think it would be useful for future CLIMFILL users to give more guidance on the methods to use in each step of the algorithm. Table 2 offers many possible choices, but in addition some recommendations would be needed on when to use which method and why. Also, the selection of employed variables is important as their inter-relations are a key source for the gap reconstructions, so also some additional advice on this would be helpful.
- AR: We thank Rene Orth for this helpful feedback. As for the selection of employed variables, please see our reply to the next comment. As for the possible choices, we have adapted the manuscript to give CLIMFILL users more guidance on how to apply the framework: We acknowledge that testing all the possible choices in Table 2 and provide recommendations in which settings they should be used is out of scope for this work. We therefore have removed the Table 2. To nevertheless make the framework easily applicable to users, we have detailed the hyper-parameter choices and their specific reasoning in Appendix Table A3.

We believe this way with the settings described in this manuscript CLIMFILL is easily adaptable to the individual gap-filling needs of users.

- RC: I think that the feature selection is a bit arbitrary and dependent on expert knowledge. To somewhat address this issue, maybe several features could be used by default, such as the 34 features used in the presented example and maybe even additional time lags and windows. Then, the random forest model can be employed to rank the features by their importance (e.g. using SHAP value importance) to make a more informed decision on the useful features. Finally, the gap-filling could be re-run with only retaining relevant features.
- AR: This is a very helpful comment. Anonymous Referee #1 also suggested adding more time lags and windows, especially in the future. The initial choice indeed was indeed guided by expert knowledge. For the revised version we have now added time lags and windows such that they are symmetric for past and future. Additionally, we have tried 3 flavors of feature selection which we discuss in the first part of the results and then settle on the best performing feature subset for the rest of the results. The 3 flavours are:
  - only the four variables that have missing values
  - the four variables plus all embedded features
  - the four variables plus all embedded features plus all constant maps

Please note the respective changes in the text:

The above procedure thus results in a set of 34 features: The four variables, the six embedded features of each of the four variables, totalling in 24 embedded features, the six maps and latitude, longitude and time information. All data are standardized to have zero mean and a standard deviation of one. We perform feature selection experiments (only the four variables, all embedded features, all embedded and constant features) to find the most descriptive subset of these 34 features, which we then use for computing the results.

- RC: There is advanced statistical and data science language used across the manuscript and I recommend to clarify this with additional information to allow a broader geoscientific audience to follow this manuscript. Please see my respective suggestions in the specific comments below.
- AR: Thank you for pointing this out. We have done our best to remove or introduce data science language where possible, starting with your specific suggestions below.
- **RC:** *line 2: estimates for what?*
- AR: The sentence has been changed to make this more clear:

Their abundance and often complex patterns can be a barrier for combining different observational datasets and may cause biased estimates of derived statistics.

- RC: line 5: remove "up"
- AR: the wording is changed accordingly.

Here we propose CLIMFILL (CLIMate data gap-FILL), a multivariate gap-filling procedure that builds up upon simple interpolation by additionally applying a statistical imputation method which is designed to account for dependence across variables.

- RC: line 7: I agree that technically the algorithm does not require a gap-free donor variable; however if all variables have gaps at the same time and if this period is longer, then the final gap-fill estimate will naturally have a low quality
- AR: Yes, it is true that if all variables have gaps at the same time this will likely affect the quality of the estimate. Note, however, that this does not have to result in poor overall performance in case of a good first guess in step 1. In other words, if all variables have gaps at the same time and if this period is longer, the final gap fill estimate will be the gap fill estimate of the first step, the spatial interpolation, which is a state-of-the-art interpolation method.
- RC: line 15: "profit", maybe rephrase as "are improved by"
- AR: The text has changed, therefore this comment is not applicable anymore.
- RC: lines 45, 144 & Table 1: Jung et al., (2019) and O. and Orth, (2021)are relevant studies in this context and could be mentioned here
- AR: We thank the reviewer for pointing us towards this very relevant literature and have included them in our introduction:

Several data products gap-fill one or more observations to a spatially or temporally complete data sets using auxiliary vari- ables (Huffmann et al., 2019; Brocca et al., 2014) or estimate variables that are only observed through sparse station networks 55 through statistical up-scaling (O. and Orth, 2021; Zhang et al., 2021; Ghiggi et al., 2019; Jung et al., 2019; Martens et al., 2017; Gudmundsson and Seneviratne, 2015; Jung et al., 2011, 2009).

**RC:** *line 46: please clarify "scale somewhere between"**

AR: The text has changed, therefore this comment is not applicable anymore.

**RC: line 84: please clarify "difficult observational record"**

AR: We have changed the wording from "difficult" to "fragmented" in coherence with line 28, 43 and 51 of the originally submitted manuscript

For example, in the state-of-the-art atmospheric reanalysis product ERA-5 the difficult fragmented observational record of soil moisture is used only sparsely (Hersbach et al., 2020), although the added value of assimilating remote sensing soil moisture has been shown for weather forecast models (Zhan et al., 2016) and flood forecasting (Brocca et al., 2014; Sahoo et al., 2013).

**RC:** lines 108/109 and 111 are in contrast to each other**

AR: A good point. We have removed the sentence "All these three missingness patterns can be observed in earth observation data:" such that consistency is ensured.

. All these three missingness patterns can be observed in Earth observation data

- **RC:** *line 151: this is unclear, please rephrase*
- AR: *The text has changed, therefore this comment is not applicable anymore.*
- RC: line 154: another" should be "other" I guess
- AR: The text has changed, therefore this comment is not applicable anymore.
- RC: Table 2, caption: "other" should be "another" I guess
- AR: Table 2 has been omitted. This comment is therefore no longer applicable.
- RC: Table 2, right column: "or more complex interpolation methods", "Guided by ...", these are not exactly examples as the column title suggests
- AR: Table 2 has been omitted. This comment is therefore no longer applicable.
- RC: line 170: remove "on"
- AR: The wording is changed accordingly.

Finally, Sect. 4 discusses the results and provides a conclusion and an outlook on for possible future work.

**RC: line 171: feels a bit random which letters are capitalized here and which are not**

AR: The section title is changed to:

CLIMFILL v1: A Generalised Framework for Infilling Missing Values in Multivariate spatio-temporal geoscientific Data

**RC:** *line 173: "the highly structured nature", please explain**

AR: The sentence is changed to:

We aim for a multivariate gap-filling framework that exploits the highly structured naturespatial, temporal and cross-variable dependence structure of Earth system observations to produce estimates for missing values.

**RC: Figure 2, caption: The framework is divided into four steps, not three.**

AR: The sentence is changed to:

Overview on the structure of the gap-filling framework. The framework is divided into three four steps.

**RC:** *line 178: Abbreviation CLIMFILL is mentioned earlier and should be explained at the first occasion**

AR: *We remove the explanation of the abbreviation here:*

The framework CLIMFILL (CLIMate data gap-FILL) works mutually, i.e. information available in each of the variables is used for filling the gaps of all the other variables.

and insert it in the abstract at the first occurrence:

Here we propose CLIMFILL (CLIMate data gap-FILL), a multivariate gap-filling procedure that builds up upon simple interpolation by additionally applying a statistical imputation method which is designed to account for dependence across variables.

**RC: line 181: please clarify "correlation structure"**

AR: We replaced "correlation structure" with "dependence structure" to ensure consistency across the text. (for example, lines 189, 366, 450,...)

With this design we implicitly assume that if one variable is not observed at a certain space-time point, a subset of the other variables might be observed and can reconstruct the missing value while conserving the correlation dependence structure among all variables.

**RC: lines 203, 311: please clarify "constant"**

AR: We replace the word "constant" with "time-indepentent in the first instance, and remove the word "constant" from "constant maps" in the second instance because "maps" already implies the fact that these features do not change with time.

For example, gap-free constant maps of time-independent maps describing properties of the land surface such as topography or land cover can be included.

Constant maps Maps of altitude, topographic complexity, land cover class and land cover height from ERA5 as well as latitude, longitude and time are added to the list of features and copied for each time step.

**RC:** *line 216: quotation marks not needed**

AR: The sentence is changed accordingly.

Earth observations often inform about time dependent processes like seasonal effects, weather persistence or soil moisture memory effects that act from daily to monthly or subseasonal time scales (Nicolai-Shaw et al., 2016).

**RC: lines 229: please clarify "stabilising the results"**

AR: Since the Algorithm has been changed such that only one clustering is done, this paragraph is removed.

For stabilising the results and to reduce the risk of discontinuities at the cluster edges, the clustering procedure is repeated E times with different numbers of terminal clusters on copies of the data  $\mathbf{X}^{(1)}, \dots, \mathbf{X}^{(E)}$ .

**RC: line 231: please clarify "terminal clusters"**

For stabilising the results and to reduce the risk of discontinuities at the cluster edges, the clustering procedure is repeated E times with different numbers of terminal clusters on copies of the data  $\mathbf{X}^{(1)}, \dots, \mathbf{X}^{(E)}$ .

AR: Since the Algorithm has been changed such that only one clustering is done, this paragraph is removed.

**RC: line 243: I think this should be "to overwrite the former estimates"**

AR: The wording is changed accordingly.

(1) all data points where  $\mathbf{y}_{v}^{(e,k)}$  was originally observed are used to fit the supervised learning method  $\mathbf{y}_{v,o}^{(e,k)} = f(\mathbf{X}_{-v,o}^{(e,k)})$  and (2) all data points where  $\mathbf{y}_{v}^{(e,k)}$  was missing  $\mathbf{y}_{v,m}^{(e,k)}$  are predicted from the fitted function and to overwrite the former estimates:

**RC: lines 250/251: "learns different weights", please clarify**

AR: We replace "weights" with "model parameters" for clarification.

Note that the framework is set up such that each cluster applies the same supervised learning method but learns different weights model parameters.

**RC: Figure 3, caption: replace "substracting" with "subtracting"**

AR: The text has changed, therefore this comment is not applicable anymore.

**RC:** *line 272: How are deserts defined and detected?**

AR: We add the definition for deserts for clarification:

Permanently glaciated areas and deserts (defined as areas with less 50 mm average yearly precipitation in the years 2003-2012) are masked.

**RC: line 311: It should be 4 and not 3 additional features I guess?**

- AR: This paragraph has been rewritten as a response to comments on feature engineering above. This comment is therefore not applicable anymore.
- RC: line 314: please clarify "non-normality"
- AR: the log-scaling of the preciptation has been removed, therefore this sentence has been removed.

Furthermore, precipitation is divided into a log-scaled precipitation-amount variable and a binary precipitation-event variable to treat its inherent non-normality.

**RC:** *line 316: How does this add up to 34?**

AR: As a response to comments above, the number of features has been changed. The text now explains the number of features here:

The above procedure thus results in a set of 34 features: The four variables, the six embedded features of each of the four variables, totalling in 24 embedded features, the six maps and latitude, longitude and time information.

**RC: line 319: "respectively" should be added after "clusters" I guess**

- AR: the clustering algorithm has been changed, therefore this comment is not applicable anymore.
- **RC:** *line 326: I wonder if and how different spatial resolutions can affect the accuracy of the gap filling, it would be great if the authors could shortly discuss this.*
- AR: We have changed the cross-validation procedure, which is now run on the original resolution. Therefore this comment is not applicable anymore.
- RC: line 326: "where one fold is one year", please clarify
- AR: We have changed the cross-validation procedure, which is now run on the original resolution. Therefore this comment is not applicable anymore.
- **RC:** Figure 7, caption: what is "CLIMPUTE-RF"?
- AR: Thanks for spotting this typo. We change it to:

Map of B-distance of univariate interpolation (a) and CLIMFILL-RF (b) as well as B-distance per land cover type (c) and altitude (d) for interpolation gap-fill and CLIMPUTE-RF-CLIMFILL gap-fill in real missingness case.

**RC: line 351: please clarify "det"**

- AR: The B-distance has been replaced, therefore this paragraph has been rewritten and this comment is not applicable anymore.
- RC: Figure 8, caption: sentences should not end with "with" and "create".
- AR: Thank you. We have changed the caption to incorporate the comments:

Comparison of (a) artificial (a) random and (b) swaths-only missingness and (c) missingness in the real data in example snapshot of ERA5 ground temperature on 1st of August 2003 with. 2003. Random missingness was created by randomly sampling without replacement from the pool of all gridpoints on land at all timesteps in the desired fraction of missing values. In swaths-only missingness we create long ellipses centered around the equator to simulate characteristic satellite swath missingness patterns. Note that the two missingness patterns are not exactly the same for each day and variable to allow for mutual learning.

- RC: line 361: "This" should be added before "leads".
- AR: The B-distance has been replaced, therefore this paragraph has been rewritten and this comment is not applicable anymore.
- **RC:** line 367, section 3.4: I very much like the idea of studying the performance of the gap-filling across missingness patterns and different severity of the gaps.
- AR: We thank the reviewer very much for this feedback.
- RC: Figure 10, caption: the B-distance is not actually displayed in this figure
- AR: Thank you for spotting this. We have changed the sentence to:

Median performance of gap-filling with CLIMFILL-RF on different missingness patterns and fractions of missingness expressed in three metrics: pearson correlation , and root mean square error (RMSE) and B-distance (for more detail see text) per variable

**RC:** *line 373: How exactly are the satellite swaths imitated?**

AR: We have added an explanation in caption of Figure 8 as a response to Referees #1 and #2:

In swaths-only missingness we create long ellipses centered around the equator to simulate characteristic satellite swath missingness patterns.

**RC:** *line 401: I do not quite understand the point on the bias correction.**

AR: Thank you for this comment. As a reaction to this comment and the one of Referee #1 regarding bias correction, we have decided to remove the mentioning of bias correction in the manuscript. It is not straight-forward to apply and this would be out of the scope of this manuscript.

- **RC:** *line 427: similar in "remotely sensed" data but underestimated in "satellite observations", this should be the same thing?*
- AR: The text has changed, therefore this comment is not applicable anymore.
- **RC:** Figure 2: The figure is rather small now and should be enlarged to make it easier to see all details.
- AR: We have enlarged the Figure.
- RC: Figure 4: The months axis should not go to 12.5
- AR: The figure is changed accordingly.

---

## Author Response (AR2)

**Reply to Reviewer Comments to the Manuscript**

**CLIMFILL v0.9: A Framework for Intelligently Gap-filling Earth Observations**

Verena Bessenbacher, Sonia I. Seneviratne, Lukas Gudmundsson
*Geophysical Model Development,* `doi:10.5194/gmd-2021-164`
* * *
**RC:** *Reviewer Comment*,     AR: *Authors response*,     ☐ Manuscript text

**1. Letter to the editor**

Dear Prof. Tomomichi Kato,

Please find enclosed the second revised version of the article entitled "CLIMFILL v0.9: A Framework for Intelligently Gap filling Earth Observations" (Paper GMD-2021-164). In the revised version, we have

- implemented the technical corrections provided by Anonymous Referee #1.

- revised the manuscript according the revisions suggested by Anonymous Referee #2 in all the cases where the comments of Anonymous Referee #2 were still applicable to the current version of the manuscript, as we have previously agreed via Email correspondence.

- updated the results and Figures taking into account the fact that we were able to run CLIMFILL on a larger time period whilst the revisions took place. One major drawback of the new methodology introduced in the revised version was that the computational expensiveness grew such that we had to restrict ourselves to gapfilling only a year (2003) of data. Now, since we had some time in between revisions, we were able to run CLIMFILL for the time period 2003-2020, and included the results. This also leads to adding a new Figure (Figure 12) that showcases the interannual variability, which was previously impossible since results were only available for one year.

- thoroughly revised the text to improve readability and comprehension. We replaced the RMSE with pearson correlation in Figure 11 and 12 to improve intuitive understanding of CLIMFILLs advantages and shortcomings.

These minor changes can be viewed in the accompanied difference-document. We are confident that the above-mentioned revisions of the paper together with the updated supporting information have increased the value of the submitted manuscript. We are looking forward to your reply and the reply of the Referees, if applicable.

Yours Sincerely,

Verena Bessenbacher (on behalf of all co-authors)

**2. Anonymous Referee #1**

**RC:** *The authors have substantially revised their manuscript and entirely revisited the methodology. I congratulate them for the thorough work and I believe the manuscript has improved and in now publishable.*

*I only have a few minor comments that can be addressed without a further round of revisions:*

**AR:** *We thank Anonymous Referee #1 for this positive feedback*

**RC:** *Section 2.1, step 1: I agree an approximation might be needed for computational reasons, but it would be good to mention that a consequence is that uncertainty in the interpolation cannot be estimated.*

**AR:** *We agree that with this the multistep process in step 1 the uncertainty of the interpolation cannot be estimated. Firstly, thin-plate-spline interpolation does not offer uncertainty estimates, and secondly it is unclear how the uncertainties of the different iterations could be aggregated. Further research would be needed to answer this question. We have added the following sentence to this section:*

> The interpolation of the daily anomalies follows Das et al (2018), who suggest reducing complexity of kriging/Gaussian Process regression by repeated interpolations on random sub-samples of all available data points and averaging the resulting estimates. In particular, the missing values in the anomalies are estimated by randomly selecting 1000 observed points per month over which the interpolation is calculated. This is repeated five times and the mean of all interpolations for each missing point is taken as the gap-fill estimate. As a consequence of these adaptations, the interpolation step becomes computationally feasible, but the uncertainty of the interpolation cannot be estimated. Finally, monthly maps and anomalies are summed up to form the inital gap-fill estimate from step 1.

**RC:** *l. 302: There is a small mistake here: In the JS divergence, zero means that both distributions are identical.*

**AR:** *Thanks for spotting this error. We have corrected the sentence, it reads now:*

> This measure compares the distance between two multivariate distributions, where a value of zero means that both distributions are identical, and one indicates that the distributions are not overlapping.

**RC:** *Caption of figure 11: Please clarify that the top panel indicates RMSE while the bottom panel indicates physical values.*

**AR:** *Figure 11 now shows pearson correlation and not RMSE. To clarify this and reply to this comment, We have changed the caption to:*

>  Pearson correlation of regionally averaged mean seasonal cycles between CLIMFILL and the original ERA-5 data over IPCC reference regions (AR6 regions, Iturbide et al. 2020) (top panel maps). Regionally averaged mean seasonal cycle of the physical values over selected regions in original ERA-5 data, satellite-observed ERA-5 data and  gap filled CLIMFILL data (bottom panels). The selected regions  show exemplary advantages and problems of the framework, see text. For all other AR6 regions see  Appendix Fig. A2.

**RC:** *Reference to Allard (2013): This is a citation to the review of a book. Please check whether you prefer to refer the book itself.*

**AR:** *We have replaced the reference to Allard (2013) and Chilès and Delfiner (1999) with the reference to the most recent edition of the respective book:*

> In the geoscientific literature, among the most commonly used approaches for estimating unobserved points are spatial and temporal interpolation methods, including nearest neighbour regression as well as kriging and derivatives thereof (Liu et al., 2018; Cowtan and Way, 2014; Haylock et al., 2008; Cressie et al., 2006, for an overview see  Cressie and Wikle 2015; Chiles and Delfiner 2012)).

*where the full reference is: Chiles, J.-P. and Delfiner, P.: Geostatistics: modeling spatial uncertainty, Wiley series in probability and statistics, Wiley, Hoboken, N.J, 2nd ed edn., 2012.*

**3.  Anonymous Referee #3**

**RC:** *The authors proposed a method to fill missing values in gridded hydrometeorological data. The key idea is to estimate missing data for one variable (e.g. soil moisture) by using some other independent variables (e.g. precipitation, ground temperature). The authors applied their method and claimed that it performed better than simple interpolation.*

*Geoscientists are always bothered by missing data. Efficient and reliable methods of filling in missing data are always eagerly expected. Although the method by authors could be potentially promising, I am not totally convinced so in the current form of manuscript due to a simple single reason. I guess (because I found only very short qualitative description for this in lines 474-482) that the performance of data reconstruction is quite condition-dependent (e.g. the parameters, quality of original data, period and location of validation, etc). I would like to see discussion how sensitive the performance is to such conditions. I totally understand this is a demanding request, but at least I wish to see at least a relevant and detailed discussion in the manuscript.*

**AR:** *We thank Referee #3 for the feedback and the detailed comments below. We note, however, that the comments seem to refer to the originally submitted version of the manuscript, and not the resubmitted one which was under discussion. In discussion with the Topical Editor it was decided that we reply to the comments wherever they are still relevant to the current version of the manuscript.*

*Specific comments*

**RC:** *Line 216 "soil moisture values at this point in a 3-month backward window (s=90 days) from the current date (l = 0days), corresponding to previous work indicating the soil moisture memory effect acts on "monthly to subseasonal time scales": If I understood correctly, the parameter 90 and 0 was more or less "subjectively" decided by the authors. Should the users of your method need to conduct sensitivity tests for these parameters? If yes, is it implementable (i.e. parameter combinations can be easily exploded). If not, what is the rationale?*

**AR:** *The chosen parameters were decided by taking into account previous work with embedded features (Ghiggi et al, 2019, Gudmundsson et al, 2015) and are intended to reflect the timescales in which soil moisture memory effects are predominantly taking place (see e. g. Nicolai-Shaw et al. 2016, full citation in paper). Ideally,*

*any parameter that is set or needs to be set within the CLIMFILL framework would be determined using the Cross-Validation procedure described in Section 2.4. However, we discuss in Section 2.4 that some of the parameters are set such that computational expense still is within manageble limits. Similarly, we do currently not recommend sensitivity tests for the parameters mentioned here, since as already noted parameter combinations can easily explode and running many instances of CLIMFILL to test different parameter combinations is too expensive to be feasible.*

RC:  ***Line 237 "the algorithm repeatedly iterates over the variables until convergence is reached": The term "convergence" needs definition. Convergence toward what?***

AR:  *In the current version of the manuscript, we have defined convergence in line 232-234:*

> The algorithm is stopped (stopping criterion) once the change in the estimates for the missing values is small between iterations (convergence) or a maximum number of iterations is reached (early stopping).

RC:  ***Lines 301-327 In this part, numerous parameters appear without showing concrete background (e.g 3 d running mean, 5-pixel side length, embedded features for (s=7,l=0), (s=23, l=7), (s=150, l=30), (S=7, l=-7)). First, how these variables were determined? Second, how sensitive are these settings to the results of gap filling? Third, the authors claim that these are from the domain knowledge. Then, if one intendedly set counterfactual parameters (parameters against the domain knowledge), will the performance deteriorate? I am asking the last question because sometimes machine learning is so powerful that any data can be "predicted" by problematic assumption or data.***

AR:  *The mentioned parameters all relate to the first step of the framework (interpolation step), which has been replaced by a Gaussian Process - based gapfilling in the revised version of the manuscript, such that none of the mentioned parameters are used anymore.*

RC:  ***Line 404 "multivariate property of land climate interactions that is currently underestimated in satellite data": What does it mean by underestimation in property? I found more detailed explanation in lines 426-428, but anyway it is a bit hard to read.***

AR:  *This paragraph and Figure 12 which it relates to is removed in the revised version of the manuscript.*

RC:  ***Line 411 Figure 11: If I understood correctly, the difference between the "Original ERA5 data" and "Satellite-observable ERA 5 data" is mainly sourced from the grid cells to include for areal mean calculation. First, is this correct? If this is the case, the discrepancy in lines make sense because these are looking at different grid cells in each region. Then I am wondering why terrestrial water storage (TWS) is always perfectly matches between two. Is this because TWS has wide satellite-observation coverage hence the two lines are virtually the same? Is this more or less the case for ground temperature except snow free period? If all above is the case, I would say that displaying the results for TWS and ground temperature is misleading and confusing because they must well overlap.***

AR:  *This is correct: the difference between the two is that unobserved grid cells are missing in one, and not missing in the other. In Figure 11 in both versions of the mansucript, TWS does not always perfectly match between original and satellite observed, there are slight differences (see Figures 11 and 12 and Appendix Figures A1 and A2). These differences are small because only a small amount of values (15 %, see Figure 5) are missing for TWS.*

RC:  ***Line 419 "precipitation and terrestrial water storage estimates show little change": I think precipitation shows change.***

AR:     *This part is formulated differently in the new version:*

> The most difficult case is precipitation. Precipitation estimates are only slightly improved with CLIMFILL compared to initial interpolation. Precipitation is influenced by several processes that are not captured within the four selected variables. For example, frontal rain patterns are mostly not explained by land surface properties but are governed by large scale circulation. This is a challenging case and could still furthermore be improved, for example by adding wind patterns to capture more synoptic features.

**RC:** *Figure A1: The figure caption says that "all combinations of variables", but the combination of soil moisture and ground temperature seems missing. Is there any reason for this?*

AR:     *We agree that the caption is misleading. We have changed it to:*

> Improvement of multivariate distribution with CLIMFILL gap filling: 2D-histogram  for all other combinations of variables  (apart from the one already shown in Figure 7) for the original ERA-5 data,  satellite-observable ERA-5 data, the Interpolation gapfill and CLIMFILL gapfill.

---

## Author Response (AR3)

**CLIMFILL v0.9: A Framework for Intelligently Gap-filling Earth Observations**

Verena Bessenbacher, Sonia I. Seneviratne, Lukas Gudmundsson
*Geophysical Model Development,* `doi:10.5194/gmd-2021-164`
* * *
Dear Editorial support team,

We are very happy to learn that our manuscript has been accepted for publication at GMD. Please find enclosed the final revised version of the article entitled "CLIMFILL v0.9: A Framework for Intelligently Gap filling Earth Observations" (Paper GMD-2021-164). Please note:

- Figure 13 is created by the authors (case 1 in your discussion of such cases in your submission guidelines, https://www.geoscientific-model-development.net/submission.html), therefore no additional credit needs to be added in the figure caption.

- minor changes in the text (e.g. change "Figure 4 to Fig. 4") that do not change content have been made to comply with your submission guidelines.

- We realised in Figure 6 (a) for Exp D. we have in the last submitted version not used the newest data. Therefore the boxplot in Figure 6 (a) for Exp. D is located slightly differently now, and we have deleted the accompanying sentence in the text:

> Overall, the JS-distance is lower for CLIMFILL than for interpolation globally (Fig. **??** (a)) for experiment  D. Including all variables shows overall the best results.

Please note that this does not change any results or content in any way, but corrects a mistake that we just have discovered.

If any questions arise, please do not hesitate to contact us. Please however note that the corresponding author (Verena Bessenbacher) will be on a scheduled sick leave following a minor surgery from 4th of May approx. 3 weeks. Therefore please also send any questions and communication also to lukas.gudmundsson@env.ethz.ch. Thank you very much for your understanding.

Yours Sincerely,

Verena Bessenbacher (on behalf of all co-authors)